# Learning Rule-Induced Subgraph Representations for Inductive Relation Prediction

**Tianyu Liu**[1]    **Qitan Lv**[1]    **Jie Wang**[1,2*]    **Shuling Yang**[1]    **Hanzhu Chen**[1]

[1]University of Science and Technology of China
[2]Institute of Artificial Intelligence, Hefei Comprehensive National Science Center
`{tianyu_liu, qitanlv, slyang0916, chenhz}@mail.ustc.edu.cn`
`{jiewangx}@ustc.edu.cn`

## Abstract

Inductive relation prediction (IRP)—where entities can be different during training and inference—has shown great power for completing evolving knowledge graphs. Existing works mainly focus on using graph neural networks (GNNs) to learn the representation of the subgraph induced from the target link, which can be seen as an implicit rule-mining process to measure the plausibility of the target link. However, these methods cannot differentiate the target link and other links during message passing, hence the final subgraph representation will contain irrelevant rule information to the target link, which reduces the reasoning performance and severely hinders the applications for real-world scenarios. To tackle this problem, we propose a novel *single-source edge-wise* GNN model to learn the **R**ule-induc**E**d **S**ubgraph represen**T**ations (**REST**), which encodes relevant rules and eliminates irrelevant rules within the subgraph. Specifically, we propose a *single-source* initialization approach to initialize edge features only for the target link, which guarantees the relevance of mined rules and target link. Then we propose several RNN-based functions for *edge-wise* message passing to model the sequential property of mined rules. REST is a simple and effective approach with theoretical support to learn the *rule-induced subgraph representation*. Moreover, REST does not need node labeling, which significantly accelerates the subgraph preprocessing time by up to **11.66×**. Experiments on inductive relation prediction benchmarks demonstrate the effectiveness of our REST[2].

## 1 Introduction

Knowledge graphs are a collection of factual triples about human knowledge. In recent years, knowledge graphs have been successfully applied in various fileds, such as natural language processing [1], question answering [2] and recommendation systems [3].

However, due to issues such as privacy concerns or data collection costs, many real-world knowledge graphs are far from completion. Moreover, knowledge graphs are continuously evolving with new entities or triples emerging. This dynamic change causes even the large-scale knowledge graphs, e.g., Freebase [4], Wikidata [5] and YAGO3 [6], to still suffer from incompleteness. Most existing knowledge graph completion models, e.g., RotatE [7], R-GCN [8], suffer from handling emerging new entities as they require test entities to be observed in training time. Therefore, inductive relation prediction, which aims at predicting missing links in evolving knowledge graphs, has attracted extensive attention[9, 10].

---

*The Corresponding Author.
[2]Our code is available at `https://github.com/smart-lty/REST`

The key idea of inductive relation prediction on knowledge graphs is to learn *logical rules*, which can capture co-occurrence patterns between relations in an entity-independent manner and can thus naturally generalize to unseen entities [11, 12]. Some existing models, e.g., AMIE+[13], Neural LP[14], explicitly mine logical rules for inductive relation prediction with good interpretability[14], while their performances are limited due to the large searching space and discrete optimization[15, 16]. Recently, some **subgraph-based** methods, e.g., GraIL[11], TACT[12], CoMPILE[17], have been proposed to implicitly mine logical rules by reasoning over the subgraph induced from the target link.

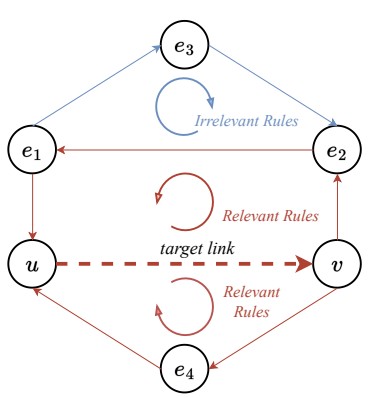

Figure 1: Relevant and irrelevant rules.

However, there are still some irrelevant rules[18] within the subgraph. Considering the rule body and the rule head as a cycle[19], the relevant rules are cycles that pass the target link. As illustrated in Figure 1, $u \rightarrow v \rightarrow e_4 \rightarrow u$ and $u \rightarrow v \rightarrow e_2 \rightarrow e_1 \rightarrow u$ are relevant rules as they pass the target link, while $e_1 \rightarrow e_3 \rightarrow e_2 \rightarrow e_1$ are irrelevant rules as they do not contain the target link. Existing methods cannot differentiate the target link and other links during message passing. Thus, they will mine plenty of irrelevant rules and encode them into the final subgraph representation, which makes the model prone to over-fitting and severely hinders reasoning performance.

In this paper, we propose a novel **single-source edge-wise** GNN model to learn the **R**ule-induc**E**d **S**ubgraph representations (**REST**), which encodes relevant rules and eliminates irrelevant rules within the subgraph. Specifically, we observe that the information flow originating from a unique edge and returning to this edge will form a cycle automatically. Consequently, the information flow originating from the target link can encode the relevant logical rules. Inspired by this observation, we propose a **single-source** initialization approach to assign a nonzero initial embedding for the target link according to its relation and zero embeddings for other links. Then we propose several RNN-based functions for **edge-wise** message passing to model the sequential property of mined rules. Finally, we use the representation of the target link as the final subgraph representation.

We theoretically show that with appropriate message passing functions, REST can learn the rule-induced subgraph representation for reasoning. Notably, REST avoids the heavy burden of node labeling in subgraph preprocessing, which significantly accelerates the time of subgraph preprocessing by up to **11.66×**. Experiments on inductive relation prediction benchmarks demonstrate the effectiveness of our REST.

## 2   Related Work

Existing works for IRP can be mainly categorized into **rule-based** methods and **subgraph-based** methods. While rule-based methods explicitly learn logical rules in knowledge graphs, subgraph-based methods implicitly mine logical rules by learning the representation of subgraphs. Moreover, we discuss some graph neural network methods that reason over the whole graph.

**Rule-based methods.**   Rule-based approaches mine logical rules that are independent of entities and describe co-occurrence patterns of relations to predict missing factual triples. Such a rule consists of a head and a body, where a head is a single atom, i.e., a fact in the form of *Relation(head entity, tail entity)*, and a body is a set of atoms. Given a head $R(y, x)$ and a body $\{B_1, B_2, \cdots, B_n\}$, there is a rule $R(y, x) \leftarrow B_1 \wedge B_2 \wedge \cdots \wedge B_n$. The rule-based methods have been studied for a long time in Inductive Logic Programming [20], yet traditional approaches face the challenges of optimization and scalability. Recently, Neural LP [14] presents an end-to-end differentiable framework that enables modern gradient-based optimization techniques to learn the structure and parameters of logical rules. DRUM[21] analyzes Neural LP from the perspective of low-rank tensor approximation and uses bidirectional RNNs to mine more accurate rules. Moreover, for automatically learning rules from large knowledge graphs, RLvLR [22] proposes an effective rule searching and pruning strategy, which shows promising results on both scalability and accuracy for link prediction. However, these explicit rule-based methods lack expressive power due to rule-based nature, and cannot scale to large knowledge graphs as well.

**Subgraph-based methods.** Subgraph-based methods extract a local subgraph around the target link and use GNNs to learn subgraph representation to predict link existence. Such a subgraph is usually induced by the neighbor nodes of the target link, which encodes the rules related to the target link. GraIL[11] is the first subgraph-based inductive relation prediction model. It defines the *enclosing subgraph* as the graph induced by all the nodes in the paths between two target nodes. After labeling all the nodes with *double radius vertex labeling*[23], it employs R-GCNs[8] to learn subgraph representation. CoMPILE[17] extracts directed enclosing subgraphs to handle the asymmetric / anti-symmetric patterns of the target link. TACT[12] converts the original enclosing subgraph into a relational correlation graph, and proposes a relational correlation network to model different correlation patterns between relations. More recently, SNRI[24] extracts enclosing subgraphs with complete neighboring relations to consider neighboring relations for reasoning. ConGLR[25] converts the original enclosing subgraph into a context graph to model relational paths. However, these methods cannot differentiate the target link and other links during message passing. Therefore, the GNNs will mine plenty of irrelevant rules for other links and encode them into subgraph representation, which reduces the accuracy of reasoning.

**Graph neural network for link prediction.** Some methods use GNNs to reason over the whole graph rather than a subgraph for inductive relation prediction. INDIGO[26] converts the KG into a node-annotated graph and fully encodes it into a GNN. NBFNet[27] generalizes Bellman-Ford algorithm and proposes a general GNN framework to learn path representation for link prediction. MorsE [28] considers transferring entity-independent meta-knowledge by GNNs. While these methods share some spirit with subgraph-based methods, they are essentially different with subgraph-based methods. These methods need to reason over the whole graph for a test example, while subgraph-based methods only need to reason over a subgraph. Meanwhile, these methods tend to predict entities rather than relations for a query, while subgraph-based methods tend to predict relations, as the subgraph only needs to be extracted once for relation prediction. As these methods benefit from a larger number of negative sampling, we do not take them into comparison.

# 3   Problem Definition

We define a training graph as $\mathcal{G}_{tr} = (\mathcal{E}_{tr}, \mathcal{R}_{tr}, \mathcal{T}_{tr})$, where $\mathcal{E}_{tr}, \mathcal{R}_{tr}$, and $\mathcal{T}_{tr} \subset \mathcal{E}_{tr} \times \mathcal{R}_{tr} \times \mathcal{E}_{tr}$ are the set of entities, relations and triples during training, respectively. We aim to train a model such that for **any** graph $\mathcal{G}' = (\mathcal{E}', \mathcal{R}', \mathcal{T}')$ whose relations are **all** seen during training (i.e., $\mathcal{R}' \subseteq \mathcal{R}_{tr}$), the model can predict missing triples in $\mathcal{G}'$, i.e., $(?, r_t, v), (u, ?, v), (u, r_t, ?)$, where $u, v \in \mathcal{E}'$ and $r_t \in \mathcal{R}'$. We denote the set of any possible entities as $\{\mathcal{E}\}$ and the set of knowledge graphs whose relation set $\mathcal{R}$ are subset of $\mathcal{R}_{tr}$ as $\{\mathcal{G}\}_{tr}$. The model we would like to train is a score function $f : \{\mathcal{G}\}_{tr} \times \{\mathcal{E}\} \times \mathcal{R}_{tr} \times \{\mathcal{E}\} \to \mathbb{R}, (\mathcal{G}', u, r_t, v) \mapsto f(\mathcal{G}', u, r_t, v)$, where $\mathcal{G}' = (\mathcal{E}', \mathcal{R}', \mathcal{T}'), \mathcal{R}' \subseteq \mathcal{R}_{tr}$ and $u, v \in \mathcal{E}'$. For a query triple $(u, r_t, ?)$, we enumerate valid candidate tail entities $v'$ and use the model to get the score $s'$ of this triple $(u, r_t, v')$. We call the query triple $(u, r_t, v)$ as *target link* and $u, v$ as *target nodes*, respectively.

# 4   Methodology

In this section, we describe the architecture of the proposed REST in detail. Following existing subgraph-based methods, we first extract a subgraph for each query triple. Then we apply **single-source initialization** and **edge-wise message passing** to update edge features iteratively. Finally, the representation of the target link is used for scoring. REST organizes the two methods in a unified framework to perform inductive relation prediction. Figure 2 gives an overview of REST.

## 4.1   Subgraph Extraction

For a query triple $(u, r_t, v)$, the subgraph around it contains the logical rules to infer this query, thus we extract a local subgraph $\mathcal{SG}_{u,r_t,v}$ to implicitly learn logical rules for reasoning. Specifically, we first compute the $k$-hop neighbors $\mathcal{N}_k(u)$ and $\mathcal{N}_k(v)$ of the target nodes $u$ and $v$. Then we define *enclosing subgraph* as the graph induced by $\mathcal{N}_k(u) \cap \mathcal{N}_k(v)$ and *unclosing subgraph* as the graph induced by $\mathcal{N}_k(u) \cup \mathcal{N}_k(v)$. Note that the subgraph extraction process of our REST omits the node labeling, as node features are unnecessary in **edge-wise** message passing, which significantly reduces the time cost of subgraph preprocessing.

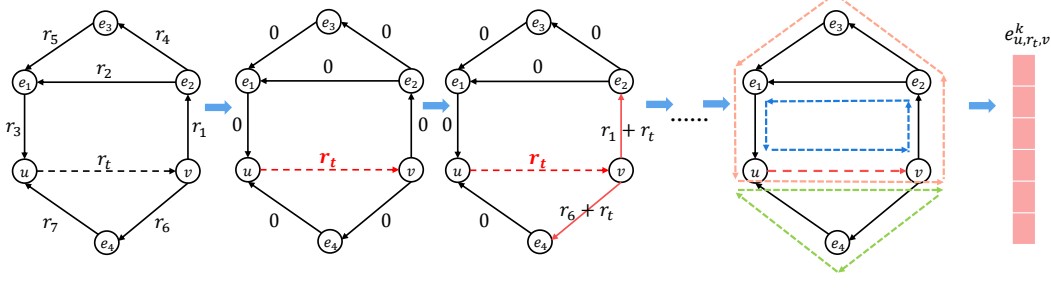

| 1. Extracted local subgraph for the target link $(u, r_t, v)$. | 2. Single-source initialization for all edges within the subgraph. | 3. Edge-wise message passing to update edge representations. Finally, we output the representation of the target link $e_{u,r_t,v}^k$ to calculate its plausibility. |
|---|---|---|

Figure 2: An overview of REST. REST organizes the *single-source* initialization method and the *edge-wise* message passing method in a unified framework to learn relevant rules representations within the subgraph for the target link. Different relevant rules are shown in different colors in part 3.

## 4.2 Single-source Initialization

Single-source initialization is a simple and effective initialization approach, which initializes a nonzero embedding to the query triple according to $r_t$ and zero embeddings for other triples. Specifically, the embeddings of links and nodes within $\mathcal{SG}_{u,r,v}$ are initialized as follows:

$$\mathbf{e}_{x,y,z}^0 = \underset{(u,r,v)}{\mathbb{1}}(x,y,z) \odot \mathbf{r}_y = \begin{cases} \mathbf{r}_y, & \text{if } (x,y,z) = (u,r_t,v) \\ \mathbf{0}, & \text{if } (x,y,z) \neq (u,r_t,v) \end{cases} \tag{1}$$

$$\mathbf{h}_v^0 = \mathbf{0} \qquad for\ \forall v \in \mathcal{E},$$

where $\mathbf{e}_{x,y,z}^0$ and $\mathbf{h}_v^0$ are the initial representation of edge $(x, y, z)$ and node $v$, respectively. $\mathbb{1}$ is the indicator function to differentiate the target link and other links. $\odot$ is Hadamard product. Note that the representation of nodes is used as temporary variables in edge-wise message passing. With this initialization approach, we ensure the relevance between mined rules and the target link.

## 4.3 Edge-wise Message Passing

After initializing all the edges and nodes, we perform edge-wise message passing to encode all relevant rules into the final subgraph representation. Specifically, each iteration of edge-wise message passing consists of three parts, (1) applying message function to every link, (2) updating node features by aggregating message and (3) updating edge features by temporary node features, which are described as follows:

$$\mathbf{m}_{x,y,z}^k = \text{MESSAGE}(\mathbf{h}_x^{k-1}, \mathbf{e}_{x,y,z}^{k-1}, \mathbf{r}_y) = (\mathbf{h}_x^{k-1} \otimes^1 \mathbf{r}_y) \uplus (\mathbf{e}_{x,y,z}^{k-1} \otimes^2 \mathbf{r}_y) \tag{2}$$

$$\mathbf{h}_z^k = \text{AGGRAGATE}(\mathbf{m}_{x,y,z}^k) = \bigoplus_{(x,y,z)\in\mathcal{T}} \mathbf{m}_{x,y,z}^k \tag{3}$$

$$\mathbf{e}_{x,y,z}^k = \text{UPDATE}(\mathbf{h}_x^k, \mathbf{e}_{x,y,z}^{k-1}) = \mathbf{h}_x^k \diamond \mathbf{e}_{x,y,z}^{k-1} \tag{4}$$

Here, $\uplus, \oplus, \diamond, \otimes^1, \otimes^2$ are binary operators which denote a function to parameterize. $\bigoplus$ denotes the large size operator of $\oplus$. $\mathbf{h}_z^k$ and $\mathbf{e}_{x,y,z}^k$ respectively represent the feature of node $z$ and link $(x, y, z)$ after $k$ iterations of edge-wise message passing. We visualize the comparison between conventional message passing framework developed by GraIL[11] and proposed edge-wise message passing framework in Figure 3.

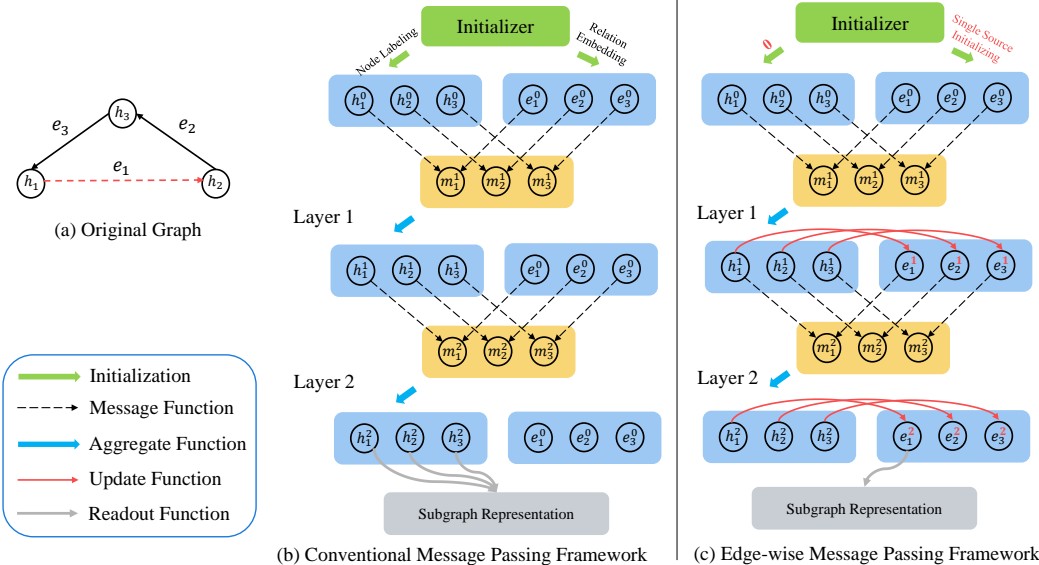

Figure 3: Comparison between conventional message passing framework developed by GraIL[11] and our REST. First, REST initializes node and edge features with single-source initialization. Then, REST employs UPDATE function to update edge features. Finally, REST directly uses the embedding of the target link as the final subgraph representation, rather than the pooling of all node embeddings.

## 4.4 RNN-based Functions

Message passing functions in existing works use order-independent binary operators such as ADD and MUL, which cannot model the sequential property of rules and lead to incorrect rules[21]. To tackle this problem, we introduce several RNN-based methods as message passing functions.

**Message Functions.** For the edge-wise message passing process, each iteration REST takes in $\mathbf{h}_x^{k-1}$, $\mathbf{e}_{x,y,z}^{k-1}$ and $\mathbf{r}_y$ to form a message. We modify GRU[29] as message function as follows:

$$
\begin{aligned}
\delta_k &= \sigma_g(\mathbf{W}_{\delta,1}^k \mathbf{r}_y \odot \mathbf{e}_{x,y,z}^{k-1} + \mathbf{W}_{\delta,2}^k \mathbf{h}_x^{k-1} + \mathbf{b}_\delta^k) \\
\gamma_k &= \sigma_g(\mathbf{W}_{\gamma,1}^k \mathbf{r}_y \odot \mathbf{e}_{x,y,z}^{k-1} + \mathbf{W}_{\gamma,2}^k \mathbf{h}_x^{k-1} + \mathbf{b}_\gamma^k) \\
c_k &= \sigma_h(\mathbf{W}_{c,1}^k \mathbf{r}_y \odot \mathbf{e}_{x,y,z}^{k-1} + \mathbf{W}_{c,2}^k (\gamma_k \odot \mathbf{h}_x^{k-1})) \\
\mathbf{m}_{x,y,z}^k &= \delta_k \odot c_k + (1 - \delta_k) \odot \mathbf{h}_x^{k-1}
\end{aligned}
\tag{5}
$$

Here, $\delta_k$ is the update gate vector, $\gamma_k$ is the reset gate vector and $c_k$ is the candidate activation vector. The operator $\odot$ denotes the Hadamard product, $\sigma_g$ denotes Sigmoid activation function and $\sigma_h$ denotes Tanh activation function. During each iteration of message passing, we only use GRU once, therefore $k$-layer message passing includes $k$ GRUs, which can model sequence with length $l \leq k$.

**Aggregate Functions.** The aggregate function aggregates messages for each node from its neighboring edges. Here, we use simplified PNA[30] to consider different types of aggregation.

$$
\begin{aligned}
\mathbf{h}_{z,1}^k &= \underset{(x,y,z)\in\mathcal{T}}{mean}(\mathbf{m}_{x,y,z}^k), \ \mathbf{h}_{z,2}^k = \underset{(x,y,z)\in\mathcal{T}}{max}(\mathbf{m}_{x,y,z}^k), \\
\mathbf{h}_{z,3}^k &= \underset{(x,y,z)\in\mathcal{T}}{min}(\mathbf{m}_{x,y,z}^k), \ \mathbf{h}_{z,4}^k = \underset{(x,y,z)\in\mathcal{T}}{std}(\mathbf{m}_{x,y,z}^k), \\
\mathbf{h}_z^k &= \mathbf{W}_{agg}^k[\mathbf{h}_{z,1}^k; \mathbf{h}_{z,2}^k; \mathbf{h}_{z,3}^k; \mathbf{h}_{z,4}^k; \mathbf{h}_z^{k-1}]
\end{aligned}
\tag{6}
$$

Here, $[;]$ denotes the concatenation of vectors, $\mathbf{W}_{agg}^k$ denotes the linear transformation matrix in the $k$-th layer.

**Update Functions.** The update function is used to update the edge feature. We propose to update the edge feature with LSTM[31]. Specifically, LSTM needs three inputs: a hidden vector, a current

input vector and a cell vector. We use $\mathbf{h}_x^k$ as the hidden vector and $\mathbf{e}_{x,y,z}^{k-1}$ as the current input vector. Moreover, we expect that each edge can *differentiate the target link* during message passing, which requires each edge to specify the query information. Therefore, we initialize each edge with another *query* feature as the cell vector. All the edges are initialized with the same query embedding $\mathbf{r}_r^q$ related to the query relation $r$.

$$\mathbf{q}_{x,y,z}^0 = \mathbf{r}_r^q \tag{7}$$

Then the update function can be described as follows:

$$\mathbf{e}_{x,y,z}^k, \mathbf{q}_{x,y,z}^k = \mathbf{LSTM}(\mathbf{e}_{x,y,z}^{k-1}, \mathbf{q}_{x,y,z}^{k-1}, \mathbf{h}_x^k) \tag{8}$$

After updating the edge feature by $k$ iterations of edge-wise message passing, we output $\mathbf{e}_{u,r,v}^k$ as the subgraph representation. Then we use a linear transformation and an activation function to get the score of the target link.

$$f(u, r_t, v) = \sigma(\mathbf{W}_s \mathbf{e}_{u,r_t,v}^k + \mathbf{b}_s) \tag{9}$$

## 5 Analysis

In this section, we theoretically analyze the effectiveness of our REST. We first define the rule-induced subgraph representation, which utilizes encoded relevant rules to infer the plausibility of the target link. Then we show that our REST is able to learn such a rule-induced subgraph representation for reasoning.

### 5.1 Rule-induced Subgraph Representation Formulation

Our rule-induced subgraph representation aims to encode all relevant rules into the subgraph representation for reasoning. Therefore, we can define the rule-induced subgraph representation as the aggregation of these relevant rules:

$$\mathcal{S}_{u,r_t,v} = \bigoplus_{c \in \mathcal{C}} \mathbf{p}_c, \tag{10}$$

where $\mathcal{C}$ denotes the set of all possible relevant rules within $\mathcal{SG}_{u,r_t,v}$ and $\mathbf{p}_c$ is the representation of a relevant rule $c$. Following the idea of Neural LP[14] to associate each relation in the rule with a weight, we model the representation of a rule as a function of its relation set. Therefore, we give the definition of rule-induced subgraph representation.

**Definition 1 (Rule-induced subgraph representation.)** *Given a subgraph $\mathcal{SG}_{u,r_t,v}$, its rule-induced subgraph representation is defined as follows:*

$$\mathcal{S}_{u,r_t,v} = \bigoplus_{i=1}^k \underbrace{\bigoplus_{(u,r_t,v)} \bigoplus_{(v,y_0,x_0)} ... \bigoplus_{(x_{i-3},y_{i-2},u)}}_{i} \alpha_{i1}\mathbf{r}_{r_t} \otimes \alpha_{i2}\mathbf{r}_{y_0} \otimes ... \otimes \alpha_{ii}\mathbf{r}_{y_{i-2}} \tag{11}$$

*where $i$ denotes the length of the cycle, $\mathbf{r}_{y_i}$ is the representation of relation $y_i$, $(x_i, y_i, x_{i-1})$ is an existing triple in $\mathcal{SG}_{u,r_t,v}$. $\{(u,r_t,v), (v,y_0,x_0), ..., (x_{i-3},y_{i-2},u)\}$ is a cycle at length $i$.*

Note that $\oplus$ and $\otimes$ denote binary aggregation functions. Intuitively, rule-induced subgraph representation captures all relevant rules within the subgraph and is expressive enough for reasoning.

### 5.2 Rule-induced Subgraph Representation Learning

Here, we show that our REST can learn such a rule-induced subgraph representation. First, we show this in a simple case.

**Theorem 1** *Single-source edge-wise GNN can learn rule-induced subgraph representation if $\uplus = +, \oplus = +, \diamond = +, \otimes^1 = \times, \otimes^2 = \times$. i.e., there exists nonzero $\alpha_{i,j}$ such that*

$$\mathbf{e}_{u,r_t,v}^k = \sum_{i=1}^k \underbrace{\sum_{(u,r_t,v)} \sum_{(v,y_0,x_0)} ... \sum_{(x_{i-3},y_{i-2},u)}}_{i} \alpha_{i1}\mathbf{r}_{r_t} \times \alpha_{i2}\mathbf{r}_{y_0} \times ... \times \alpha_{ii}\mathbf{r}_{y_{i-2}} \tag{12}$$

We prove this in Appendix A. This theorem states that our REST can learn the rule-induced subgraph representation in the basic condition. Then we generalize this theorem to a general version.

**Theorem 2** *Single-source edge-wise GNN can learn rule-induced subgraph representation if $\uplus = \oplus, \oplus = \oplus, \diamond = \oplus, \otimes^1 = \otimes, \otimes^2 = \otimes$, where $\oplus$ and $\otimes$ are binary operators that satisfy $0 \oplus a = a, 0 \otimes a = 0$. i.e., there exists nonzero $\alpha_{i,j}$ such that*

$$\mathbf{e}_{u,r_t,v}^k = \bigoplus_{i=1}^{k} \underbrace{\bigoplus_{(u,r_t,v)} \bigoplus_{(v,y_0,x_0)} \cdots \bigoplus_{(x_{i-3},y_{i-2},u)}}_{i} \alpha_{i1}\mathbf{r}_{r_t} \otimes \alpha_{i2}\mathbf{r}_{y_0} \otimes \cdots \otimes \alpha_{ii}\mathbf{r}_{y_{i-2}} \tag{13}$$

We prove this in Appendix A. Intuitively, we can get this theorem by replacing $+, \times$ with $\oplus, \otimes$. The key step to learn rule-induced subgraph representation is to ensure $\mathbf{e}_{x,y,z}^0 \neq 0$ if and only if $(x, y, z) = (u, r_t, v)$. Existing models[11, 12] do not satisfy this requirement, as they initialize both the target link and the other links with nonzero embeddings. Therefore, their final subgraph representations contain irrelevant rule terms, which leads to suboptimal results. On the contrary, we show that with appropriate message passing functions, REST learns rule-induced subgraph representation. As the rule-induced subgraph representation encodes all relevant rules within the subgraph, REST is expressive enough to infer the plausibility of any reasonable triple, while it eliminates the negative influence of irrelevant rules.

Our analysis gives some insight of IRP methods. First, eliminating noises within the extracted subgraph is crucial for subgraph-based methods. While existing methods focus on data level to extract ad-hoc subgraphs, our model proposes a simple way for denoising at the model level, i.e., single-source edge-wise message passing. Second, labeling tricks such as single-source initialization can effectively improve the model performance. Last but not least, the idea of learning links is especially important in IRP task, as links play a vital role in reasoning.

## 6 Experiments

In this section, we first introduce the experiment setup including datasets and implementation details. Then we show the main results of REST on several benchmark datasets. Finally, we conduct ablation studies, case studies and further experiments.

### 6.1 Experiment Setup

**Datasets and Implementation Details** We conduct experiments on three inductive benchmark datasets proposed by GraIL[11], which are dervied from WN18RR[32], FB15K-237[33], and NELL-995[34]. For inductive relation prediction, the training set and testing set should have no overlapping entities. Details of the datasets are summarized in Appendix B. We use PyTorch[35] and DGL[36] to implement our REST. Implementation Details of REST are summarized in Appendix C.

### 6.2 Main Results

We follow GraIL[11] to rank each test triple among 50 other randomly sampled negative triples. We report the Hits@10 metric on the benchmark datasets. Following the standard procedure in prior work [37], we use the filtered setting, which does not take any existing valid triples into account at ranking. We demonstrate the effectiveness of the proposed REST by comparing its performance with both rule-based methods including Neural LP [14], DRUM [21] and RuleN [38] and subgraph-based methods including GraIL [11], CoMPILE [17], TACT[12], SNRI[24] and ConGLR [25]. We run each experiment five times with different random seeds and report the mean results in Table 1.

From the Hits@10 results in Table 1, we make the observation that our model REST significantly outperforms existing methods on 12 versions of 3 datasets. Specifically, our REST can outperform rule-based baselines, including Neural LP, DRUM and RuleN by a large margin. And compared with existing subgraph-based methods, e.g., GraIL, CoMPILE, TACT, SNRI and ConGLR, REST has achieved average improvements of 17.89%, 9.35%, 13.76%; 16.23%, 8.18%, 13.04%; 13.58%, 8.06%, 8.96%; 10.89%, 4.55%,"-" and 5.58%, 5.82%, 5.1% on three datasets respectively. As REST

Table 1: Hits@10 results on the inductive benchmark datasets extracted from WN18RR, FB15k-237 and NELL-995. The results of Neural LP, DURM, RuleN, GraIL, CoMPILE and ConGLR are taken from the paper [25].

| | | WN18RR | | | | FB15k-237 | | | | NELL-995 | | | |
| --- | --- | --- | --- | --- | --- | --- | --- | --- | --- | --- | --- | --- | --- |
| | | v1 | v2 | v3 | v4 | v1 | v2 | v3 | v4 | v1 | v2 | v3 | v4 |
| Rule-Based | Neural LP | 74.37 | 68.93 | 46.18 | 67.13 | 52.92 | 58.94 | 52.90 | 55.88 | 40.78 | 78.73 | 82.71 | 80.58 |
| | DRUM | 74.37 | 68.93 | 46.18 | 67.13 | 52.92 | 58.73 | 52.90 | 55.88 | 19.42 | 78.55 | 82.71 | 80.58 |
| | RuleN | 80.85 | 78.23 | 53.39 | 71.59 | 49.76 | 77.82 | 87.69 | 85.60 | 53.50 | 81.75 | 77.26 | 61.35 |
| Subgraph-Based | GraIL | 82.45 | 78.68 | 58.43 | 73.41 | 64.15 | 81.80 | 82.83 | 89.29 | 59.50 | 93.25 | 91.41 | 73.19 |
| | CoMPILE | 83.60 | 79.82 | 60.69 | 75.49 | 67.64 | 82.98 | 84.67 | 87.44 | 58.38 | 93.87 | 92.77 | 75.19 |
| | TACT | 84.04 | 81.63 | 67.97 | 76.56 | 65.76 | 83.56 | 85.20 | 88.69 | 79.80 | 88.91 | 94.02 | 73.78 |
| | SNRI | 87.23 | 83.10 | 67.31 | 83.32 | 71.79 | 86.50 | 89.59 | 89.39 | - | - | - | - |
| | ConGLR | 85.64 | 92.93 | 70.74 | 92.90 | 68.29 | 85.98 | 88.61 | 89.31 | 81.07 | 94.92 | 94.36 | 81.61 |
| | REST(**ours**) | **96.28** | **94.56** | **79.50** | **94.19** | **75.12** | **91.21** | **93.06** | **96.06** | **88.00** | **94.96** | **96.79** | **92.61** |

only assigns the embedding of the target link, these improvements demonstrate the effectiveness of our REST via distilling all relevant rules within the subgraph.

## 6.3 Ablation Study

We conduct ablation studies to validate the effectiveness of proposed single-source initialization and edge-wise message passing. We show the main results of ablation studies in Table 2.

Table 2: Hits@10 ablation results on the inductive benchmark datasets. The SUM and MUL functions are the ablation for the message function GRU. The MLP function is the ablation for the update function LSTM. $\Delta$ denotes the performance decrease.

| | WN18RR | | | | FB15k-237 | | | | NELL-995 | | | |
| --- | --- | --- | --- | --- | --- | --- | --- | --- | --- | --- | --- | --- |
| | v1 | v2 | v3 | v4 | v1 | v2 | v3 | v4 | v1 | v2 | v3 | v4 |
| REST | 96.28 | 94.56 | 79.50 | 94.19 | 75.12 | 91.21 | 93.06 | 96.06 | 88.00 | 94.96 | 96.79 | 92.61 |
| Full Initialization | 92.55 | 90.70 | 68.76 | 79.49 | 71.71 | 79.29 | 89.25 | 91.22 | 83.00 | 86.13 | 94.54 | 68.26 |
| $\Delta$ | -3.73 | -3.86 | -10.74 | -14.70 | -3.41 | -11.92 | -3.81 | -4.84 | -5.00 | -8.83 | -2.25 | -24.35 |
| SUM | 93.08 | 85.03 | 69.59 | 91.39 | 64.88 | 84.30 | 89.48 | 89.96 | 81.00 | 91.39 | 96.17 | 64.57 |
| $\Delta$ | -3.20 | -9.53 | -9.91 | -2.80 | -10.24 | -6.91 | -3.58 | -6.10 | -7.00 | -3.57 | -0.62 | -28.04 |
| MUL | 85.64 | 93.19 | 56.03 | 81.04 | 63.90 | 78.24 | 85.20 | 90.66 | 69.00 | 79.20 | 93.70 | 36.11 |
| $\Delta$ | -10.64 | -1.37 | -23.47 | -13.15 | -11.22 | -12.97 | -7.86 | -5.40 | -19.00 | -15.76 | -3.09 | -56.50 |
| MLP | 95.74 | 93.65 | 78.84 | 90.69 | 71.07 | 90.25 | 92.60 | 94.94 | 83.00 | 94.12 | 96.41 | 91.38 |
| $\Delta$ | -0.54 | -0.91 | -0.66 | -3.50 | -4.05 | -0.96 | -0.46 | -1.12 | -5.00 | -0.84 | -0.38 | -1.23 |

**Single-source initialization.** Single-source initialization is vital for learning rule-induced subgraph representation. To demonstrate the effectiveness of single-source initialization, we perform another *full initialization* method as a comparison, which initializes all edges according to their relations. As illustrated in Table 2, we can find that single-source initialization is significant for capturing relevant rules for reasoning. Without single-source initialization, the performance of REST will exhibit a significant decrease, e.g., from 92.61 to 68.26 in NELL-995 v4. This result exhibits the effectiveness of single-source initialization.

**Edge-wise message passing.** To demonstrate the necessity of proposed RNN-based functions, we conduct ablation studies on various combinations of message functions, including SUM, MUL, and GRU, as well as update functions, including LSTM and MLP. These functions are defined as follows:

$$
\begin{aligned}
\mathbf{SUM} &: \mathbf{m}_{x,y,z}^k = \mathbf{h}_x^{k-1} + \mathbf{e}_{x,y,z}^{k-1} + \mathbf{r}_y \\
\mathbf{MUL} &: \mathbf{m}_{x,y,z}^k = \mathbf{h}_x^{k-1} \odot \mathbf{e}_{x,y,z}^{k-1} \odot \mathbf{r}_y \\
\mathbf{MLP} &: \mathbf{e}_{x,y,z}^k = \mathbf{W}_e[\mathbf{e}_{x,y,z}^{k-1}; \mathbf{q}_{x,y,z}^{k-1}; \mathbf{h}_x^k] \\
&\quad \mathbf{q}_{x,y,z}^k = \mathbf{W}_q[\mathbf{e}_{x,y,z}^{k-1}; \mathbf{q}_{x,y,z}^{k-1}; \mathbf{h}_x^k]
\end{aligned}
\tag{14}
$$

In general, REST benefits from RNN-based functions, as they can capture the sequential properties of rules. Using order-independent binary operators, such as ADD and MUL, leads to a decline in performance across all datasets, as they cannot differentiate correct and incorrect rules.

## 6.4 Further Experiments

**Case Study** One appealing feature of our single-source initialization and edge-wise message passing approach is the ability to interpret the significance of each relevant cycle. This interpretation provides insight into the contribution of each cycle towards the plausibility of the target link $(u, r_t, v)$. We generate all relevant rule cycles for each relation with a length of no more than 4, and input them to the REST model to obtain a score for each relevant rule cycle. We normalize these scores using sigmoid function and select the top-3 cycles with the highest scores as visualized in Table 7.

Table 3: Some relations and their top-3 relevant relations. The relations are taken from WN18RR.

| Rule Head | Rule Body | Scores |
|---|---|---|
| _hypernym | _similar_to -> _hypernym$^{-1}$ | 0.89 |
| | _also_see-> _hypernym$^{-1}$ | 0.84 |
| | _hypernym$^{-1}$->_also_see->_verb_group | 0.67 |
| _derivationally_related_form | _derivationally_related_form->_also_see | 0.82 |
| | _has_part− > _also_see | 0.76 |
| | _derivationally_related_form-> _also_see-> _has_part | 0.66 |
| _member_meronym | _has_part-> _synset_domain_topic_of | 0.77 |
| | has_part-> _synset_domain_topic_of-> _derivationally_related_form | 0.76 |
| | _has_part-> _derivationally_related_form | 0.71 |
| _synset_domain_topic_of | _has_part-> _also_see | 0.95 |
| | _synset_domain_topic_of$^{-1}$-> _similar_to | 0.91 |
| | _has_part-> _synset_domain_topic_of$^{-1}$ | 0.81 |

The results demonstrate that REST is able to learn the degree of correlation between relevant rule cycles and the target relation. As evidence for predicting the rule head, the cycle containing the "$_similar\_to \rightarrow \_hypernym^{-1}$" rule body received the highest score among all cycles that include the "$_hypernym$" relationship. This indicates that REST can effectively infer a strong correlation between "$_silimar\_to \rightarrow \_hypernym^{-1}$" and "$_hypernym$". Notably, this cycle is also human-understandable, which highlights the practical interpretability of REST.

**Subgraph Extraction Efficiency** Different from the subgraph-based methods mentioned above[11, 12, 17], REST eliminates the need for node labeling within the subgraph, which substantially improves time efficiency. We assess the time consumption involved in extracting both enclosing and unenclosing subgraphs for GraIL and REST, and present the running time results in Table 4. All the experiments are conducted on the same CPU with only single process. Our observations indicate a significant improvement of time efficiency over $11\times$ when extracting unenclosing subgraphs from the FB15k-237 dataset and over $6\times$ across all inductive datasets. This improvement demonstrates the efficiency of our REST on subgraph extraction.

Table 4: The comparison of subgraph extraction time between GraIL and REST. (Unit: Second)

| | | WN18RR | | | | FB15k-237 | | | | NELL-995 | | | |
|---|---|---|---|---|---|---|---|---|---|---|---|---|---|
| | | v1 | v2 | v3 | v4 | v1 | v2 | v3 | v4 | v1 | v2 | v3 | v4 |
| Enclosing Subgraph | GraIL | 121.77 | 537.42 | 1127.14 | 194.98 | 949.48 | 2933.04 | 8423.59 | 15089.74 | 136.55 | 1197.24 | 6112.77 | 1303.97 |
| | REST | 54.01 | 251.97 | 617.16 | 91.76 | 111.34 | 338.24 | 868.79 | 1,626.77 | 61.19 | 213.45 | 688.14 | 219.33 |
| | Efficiency | 2.25× | 2.13× | 1.83× | 2.12× | 8.53× | 8.67× | 9.70× | 9.28× | 2.23× | 5.61× | 8.88× | 5.95× |
| Unclosing Subgraph | GraIL | 127.69 | 517.94 | 1194.18 | 199.00 | 1287.35 | 4166.63 | 11499.32 | 21738.29 | 167.06 | 1611.97 | 8044.53 | 1542.82 |
| | REST | 56.27 | 260.20 | 631.71 | 95.23 | 123.36 | 386.55 | 985.81 | 1890.54 | 64.72 | 245.41 | 858.23 | 248.00 |
| | Efficiency | 2.27× | 1.99× | 1.89× | 2.09× | 10.44× | 10.78× | 11.66× | 11.50× | 2.58× | 6.57× | 9.37× | 6.22× |

## 7 Conclusion and Future Work

**Limitations.** Our REST shares the same limitation as subgraph-based methods. While subgraph-based methods are theoretically more expressive, they incur high computational costs both in training and inference. Alleviating this issue is crucial for scalability.

**Conclusion.** In this paper, we propose a novel single-source edge-wise graph neural network model called REST, which effectively mines relevant rules within subgraphs for inductive reasoning. REST consists of single-source initialization and edge-wise message passing, which is simple, effective and provable to learn rule-induced subgraph representation. Notably, REST accelerates subgraph

extraction by up to $11.66\times$, which significantly decreases the time cost of subgraph extraction. Experiments demonstrate that our proposed REST outperforms existing state-of-the-art methods on inductive relation prediction benchmarks.

**Future Work.** For future work, we target at enhancing the scalability of our REST to conduct reasoning on large-scale knowledge graphs. Moreover, REST can serve as a complementary reasoning model to help large language models conduct reasoning with promising and interpretable results. Hopefully, REST will facilitate the future development of reasoning ability.

## Acknowledgements

The authors would like to thank all the anonymous reviewers for their insightful comments. This work was supported in part by National Key R&D Program of China under contract 2022ZD0119801, National Nature Science Foundations of China grants U19B2026, U19B2044, 61836011, 62021001, and 61836006.

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

# A  Proof of Theorem 1 and 2

**Theorem 1** *Single-source edge-wise GNN can learn rule-induced subgraph representation if $\uplus = +, \oplus = +, \diamond = +, \otimes^1 = \times, \otimes^2 = \times$. i.e., there exists nonzero $\alpha_{i,j}$ such that*

$$\mathbf{e}_{u,r_t,v}^k = \sum_{i=1}^k \underbrace{\sum_{(u,r_t,v)} \sum_{(v,y_0,x_0)} \cdots \sum_{(x_{i-3},y_{i-2},u)}}_{i} \alpha_{i1}\mathbf{r}_{r_t} \times \alpha_{i2}\mathbf{r}_{y_0} \times \ldots \times \alpha_{ii}\mathbf{r}_{y_{i-2}} \tag{15}$$

*Proof*  In this case, the rule-induced subgraph representation is:

$$\mathcal{S}_{u,r_t,v} = \sum_{i=1}^k \underbrace{\sum_{(u,r_t,v)} \sum_{(v,y_0,x_0)} \cdots \sum_{(x_{i-3},y_{i-2},u)}}_{i} \alpha_{i1}\mathbf{r}_{r_t} \times \alpha_{i2}\mathbf{r}_{y_0} \times \ldots \times \alpha_{ii}\mathbf{r}_{y_{i-2}} \tag{16}$$

Then we will show that single-source edge-wise GNN can learn this rule-induced sugraph representation in **induction**.

$k = 1$. we have

$$\mathbf{e}_{u,r_t,v}^1 = \mathbf{h}_u^1 + \mathbf{r}_{r_t} = \mathbf{r}_{r_t} + \sum_{(x_0,y_0,u)\in\mathcal{T}} (\mathbf{h}_{x_0}^0 + \mathbf{e}_{x_0,y_0,u}^0) \times \mathbf{r}_{y_0}$$

Note that $\mathbf{e}_{x_0,y_0,u}^0 \neq 0$ if and only if $(x_0, y_0, u) = (u, r_t, v)$. However, this is impossible as $u \neq v$. Thus $\mathbf{e}_{u,r_t,v}^1$ satisfies the definition of rule-induced subgraph representation.

$k = 2$. we have:

$$\begin{aligned}
\mathbf{e}_{u,r_t,v}^2 = \mathbf{h}_u^2 + \mathbf{e}_{u,r_t,v}^1 &= \sum_{(x_0,y_0,u)\in\mathcal{T}} (\mathbf{h}_{x_0}^1 + \mathbf{e}_{x_0,y_0,u}^1) \times \mathbf{r}_{y_0} + \mathbf{r}_{r_t} \\
&= \sum_{(x_0,y_0,u)\in\mathcal{T}} 2\mathbf{h}_{x_0}^1 \times \mathbf{r}_{y_0} + \mathbf{r}_{r_t} \\
&= \sum_{(x_0,y_0,u)} \sum_{(x_1,y_1,x_0)} 2(\mathbf{h}_{x_1}^0 + \mathbf{e}_{x_1,y_1,x_0}^0) \times \mathbf{r}_{y_1} \times \mathbf{r}_{y_0} + \mathbf{r}_{r_t} \\
&= \sum_{(x_0,y_0,u)} \sum_{(x_1,y_1,x_0)} 2\mathbf{e}_{x_1,y_1,x_0}^0 \times \mathbf{r}_{y_1} \times \mathbf{r}_{y_0} + \mathbf{r}_{r_t}
\end{aligned} \tag{17}$$

We can find that $e_{x_1,y_1,x_0}^0 \neq 0$ if and only if $(x_1, y_1, x_0) = (u, r_t, v)$, i.e. there exists both $(u, r_t, v)$ and $(v, r_t, u)$. Obviously, $\mathbf{e}_{u,r_t,v}^2$ satisfies the definition of rule-induced subgraph representation.

Assume that this conclusion exists for $n \leq k - 1$. Now we check the $k$-th term.

$$\mathbf{e}_{u,r_t,v}^k = \mathbf{h}_u^k + \sum_{i=1}^{k-1} \underbrace{\sum_{(x_0,y_0,u)} \sum_{(x_1,y_1,x_0)} \cdots \sum_{(v,y_{i-2},x_0)}}_{i-1} \alpha_{i1}\mathbf{r}_{r_t} \otimes \alpha_{i2}\mathbf{r}_{y_{i-2}} \otimes \ldots \otimes \alpha_{ii}\mathbf{r}_{y_0} \tag{18}$$

First, we consider $\mathbf{h}_u^k$.

$$
\begin{aligned}
\mathbf{h}_u^k &= \sum_{(x_0,y_0,u)} (\mathbf{h}_{x_0}^{k-1} + \mathbf{e}_{x_0,y_0,u}^{k-1}) \times \mathbf{r}_{y_0} \\
&= \sum_{(x_0,y_0,u)} \sum_{(x_1,y_1,x_0)} (\mathbf{h}_{x_1}^{k-2} + \mathbf{e}_{x_1,y_1,x_0}^{k-2}) \times \mathbf{r}_{y_1} \times \mathbf{r}_{y_0} + \sum_{(x_0,y_0,u)} \mathbf{e}_{x_0,y_0,u}^{k-1} \times \mathbf{r}_{y_0} \\
&= \underbrace{\sum_{(x_0,y_0,u)} \sum_{(x_1,y_1,x_0)} \cdots \sum_{(x_{k-1},y_{k-1},x_{k-2})}}_{k} \mathbf{e}_{x_{k-1},y_{k-1},x_{k-2}}^{0} \times \mathbf{r}_{y_{k-1}} \times \mathbf{r}_{y_{k-2}} \times \ldots \times \mathbf{r}_{y_0} \\
&+ \underbrace{\sum_{(x_0,y_0,u)} \sum_{(x_1,y_1,x_0)} \cdots \sum_{(x_{k-2},y_{k-2},x_{k-3})}}_{k-1} \mathbf{e}_{x_{k-2},y_{k-2},x_{k-3}}^{1} \times \mathbf{r}_{y_{k-2}} \times \ldots \times \mathbf{r}_{y_0} \\
&+ \ldots \\
&+ \sum_{(x_0,y_0,u)} \mathbf{e}_{x_0,y_0,u}^{k-1} \times \mathbf{r}_{y_0}
\end{aligned}
\tag{19}
$$

Notice that $\underbrace{\sum_{(x_0,y_0,u)} \sum_{(x_1,y_1,x_0)} \cdots \sum_{(x_{k-1},y_{k-1},x_{k-2})}}_{k} \mathbf{e}_{x_{k-1},y_{k-1},x_{k-2}}^{0} \times \mathbf{r}_{y_{k-1}} \times \mathbf{r}_{y_{k-2}} \times \ldots \times \mathbf{r}_{y_0} \neq 0$ if and only if $(x_{k-1}, y_{k-1}, x_{k-2}) = (u, r_t, v)$. In this situation, this term is exactly the $k$-th term in the expression of $\mathbf{e}_{u,r_t,v}^k$. Now we want to prove that:

$$
\begin{aligned}
&\underbrace{\sum_{(x_0,y_0,u)} \sum_{(x_1,y_1,x_0)} \cdots \sum_{(x_{k-2},y_{k-2},x_{k-3})}}_{k-1} \mathbf{e}_{x_{k-2},y_{k-2},x_{k-3}}^{1} \times \mathbf{r}_{y_{k-2}} \times \ldots \times \mathbf{r}_{y_0} \\
&+ \ldots \\
&+ \sum_{(x_0,y_0,u)} \mathbf{e}_{x_0,y_0,u}^{k-1} \times \mathbf{r}_{y_0}
\end{aligned}
\tag{20}
$$

can be fused in top $k-1$ term of Equation. 16. Let's check the $j$-th term of Equation. 20.

$$\underbrace{\sum_{(x_0,y_0,u)} \sum_{(x_1,y_1,x_0)} ... \sum_{(x_{j-1},y_{j-1},x_{j-2})}}_{j} \mathbf{e}^{k-j}_{x_{j-1},y_{j-1},x_{j-2}} \times \mathbf{r}_{y_{j-1}} \times ... \times \mathbf{r}_{y_0}$$

$$= \underbrace{\sum_{(x_0,y_0,u)} \sum_{(x_1,y_1,x_0)} ... \sum_{(x_{j-1},y_{j-1},x_{j-2})}}_{j} (\mathbf{e}^{k-j-1}_{x_{j-1},y_{j-1},x_{j-2}} + \mathbf{h}^{k-j}_{x_{j-1}}) \times \mathbf{r}_{y_{j-1}} \times ... \times \mathbf{r}_{y_0}$$

$$= \underbrace{\sum_{(x_0,y_0,u)} \sum_{(x_1,y_1,x_0)} ... \sum_{(x_{j-1},y_{j-1},x_{j-2})}}_{j} (\mathbf{e}^{0}_{x_{j-1},y_{j-1},x_{j-2}} + \mathbf{h}^{k-j}_{x_{j-1}} + ... + +\mathbf{h}^{1}_{x_{j-1}}) \times \mathbf{r}_{y_{j-1}} \times ... \times \mathbf{r}_{y_0}$$

$$= \underbrace{\sum_{(x_0,y_0,u)} \sum_{(x_1,y_1,x_0)} ... \sum_{(x_{j-1},y_{j-1},x_{j-2})}}_{j} \mathbf{e}^{0}_{x_{j-1},y_{j-1},x_{j-2}} \times \mathbf{r}_{y_{j-1}} \times ... \times \mathbf{r}_{y_0}$$

$$+ \underbrace{\sum_{(x_0,y_0,u)} \sum_{(x_1,y_1,x_0)} ... \sum_{(x_{j-1},y_{j-1},x_{j-2})} \sum_{(x_j,y_j,x_{j-1})}}_{j+1}$$

$$(\mathbf{h}^{k-j-1}_{x_j} + ... + \mathbf{h}^{0}_{x_j} + \mathbf{e}^{k-j-1}_{x_j,y_j,x_{j-1}} + ... + \mathbf{e}^{0}_{x_j,y_j,x_{j-1}}) \times \mathbf{r}_{y_{j-1}} \times ... \times \mathbf{r}_{y_0}$$

$$\tag{21}$$

Note that $\mathbf{e}^{0}_{x_{j-1},y_{j-1},x_{j-2}} \neq 0$ if and only if $(x_{j-1}, y_{j-1}, x_{j-2}) = (u, r_t, v)$, thus the term can be fused into the $j$-th term of Equation. 16. $\mathbf{e}^{0}_{x_j,y_j,x_{j-1}}$ can be fused into the $(j+1)$-th term and so on. Therefore, we have:

$$\underbrace{\sum_{(x_0,y_0,u)} \sum_{(x_1,y_1,x_0)} ... \sum_{(x_{j-1},y_{j-1},x_{j-2})}}_{j} \mathbf{e}^{k-j}_{x_{j-1},y_{j-1},x_{j-2}} \times \mathbf{r}_{y_{j-1}} \times ... \times \mathbf{r}_{y_0}$$

$$= \sum_{i=1}^{k} \underbrace{\sum_{(u,r_t,v)} \sum_{(v,y_0,x_0)} ... \sum_{(x_{i-3},y_{i-2},u)}}_{i} \alpha_{i1}\mathbf{r}_{r_t} \times \alpha_{i2}\mathbf{r}_{y_0} \times ... \times \alpha_{ii}\mathbf{r}_{y_{i-2}}$$

$$\tag{22}$$

There, we prove that single-source edge-wise GNN can learn rule-induced subgraph representation in this case. □

**Theorem 2** *Single-source edge-wise GNN can learn rule-induced subgraph representation if* $\uplus = \oplus, \oplus = \oplus, \diamond = \oplus, \otimes^1 = \otimes, \otimes^2 = \otimes$, *where* $\oplus$ *and* $\otimes$ *are binary operators that satisfy* $0 \oplus a = a, 0 \otimes a = 0$. *i.e., there exists nonzero* $\alpha_{i,j}$ *such that*

$$\mathbf{e}^{k}_{u,r_t,v} = \bigoplus_{i=1}^{k} \underbrace{\bigoplus_{(u,r_t,v)} \bigoplus_{(v,y_0,x_0)} ... \bigoplus_{(x_{i-3},y_{i-2},u)}}_{i} \alpha_{i1}\mathbf{r}_{r_t} \otimes \alpha_{i2}\mathbf{r}_{y_0} \otimes ... \otimes \alpha_{ii}\mathbf{r}_{y_{i-2}} \tag{23}$$

*Proof* Without loss of generality, we can replace $+$ with $\oplus$ and $\times$ with $\otimes$ to represent a binary operator, then we directly get this theorem. Note that we should ensure that $\oplus$ and $\otimes$ satisfy $0 \oplus a = a, 0 \otimes a = 0$, which we use in the process of proof. □

# B Details of Datasets

We summarize the details of inductive relation prediction benchmark datasets in Table 5.

Table 5: Statistics of three inductive datasets, which contain four different versions individually. We use #E and #R and #TR to denote the number of entities, relations, and triples.

| | | WN18RR | | | FB15k-237 | | | NELL-995 | | |
| | | #R | #E | #TR | #R | #E | #TR | #R | #E | #TR |
|---|---|---|---|---|---|---|---|---|---|---|
| v1 | train | 9 | 2746 | 6678 | 183 | 2000 | 5226 | 14 | 10915 | 5540 |
| | test | 9 | 922 | 1991 | 146 | 1500 | 2404 | 14 | 225 | 1034 |
| v2 | train | 10 | 6954 | 18968 | 203 | 3000 | 12085 | 88 | 2564 | 10109 |
| | test | 10 | 2923 | 4863 | 176 | 2000 | 5092 | 79 | 4937 | 5521 |
| v3 | train | 11 | 12078 | 32150 | 218 | 4000 | 22394 | 142 | 4647 | 20117 |
| | test | 11 | 5084 | 7470 | 187 | 3000 | 9137 | 122 | 4921 | 9668 |
| v4 | train | 9 | 3861 | 9842 | 222 | 5000 | 33916 | 77 | 2092 | 9289 |
| | test | 9 | 7208 | 15157 | 204 | 3500 | 14554 | 61 | 3294 | 8520 |

## C  Implementation Details

In general, our proposed method is implemented in DGL[36] and PyTorch[35] and trained on single GPU of NVIDIA GeForce RTX 3090. We apply Adam optimizer[39] with an initial learning rate of 0.0005. Observing that batch size has little effect on the performance of the model, We adjust batch size as large as possible for different datasets to accelerate training. We use the binary cross entropy loss.The maximum number of training epochs is set to 10. During training, we add reversed edges to fully capture relevant rules. The number of hop $h$ is set to 3 which is consistent with existing subgraph-based methods. We conduct grid search to obtain optimal hyperparameters, where we search subgraph types in {enclosing, unclosing}, embedding dimensions in {16, 32}, number of GNN layers in {3, 4, 5, 6} and dropout in {0, 0.1, 0.2}. Configuration for the best performance of each dataset is given within the code.

## D  Transductive Results

The transductive results, as discussed in Section 6.2, were obtained using the same methodology as the aforementioned evaluations. Specifically, REST was trained on the training graph and tested in a similar manner. We randomly selected 10% of the links from the training graph as test links. As we can senn in Table 6, REST outperforms GraIL and RuleN significantly across all benchmarks.

Table 6: Experiments of the transductive versions of the current benchmarks.

| Model | wn_v1 | wn_v2 | wn_v3 | wn_v4 | fb_v1 | fb_v2 | fb_v3 | fb_v4 | nell_v1 | nell_v2 | nell_v3 | nell_v4 |
|---|---|---|---|---|---|---|---|---|---|---|---|---|
| GraIL | 65.59 | 69.36 | 64.63 | 67.28 | 71.93 | 86.30 | 88.95 | 91.55 | 64.08 | 86.88 | 84.19 | 82.33 |
| RuleN | 63.42 | 68.09 | 63.05 | 65.55 | 67.53 | 88.00 | 91.47 | 92.35 | 62.82 | 82.82 | 80.72 | 58.84 |
| REST | 92.02 | 90.90 | 91.59 | 91.63 | 88.01 | 93.98 | 96.43 | 97.71 | 93.62 | 95.76 | 92.79 | 92.47 |

# E   More Results of Case Study

To gain more insight of the proposed RNN-based function and single-source initialization, We conduct more experiments to obtain the rule scores on WN18RR with the ablated model without single-source and other ablated models. In the revised paper, we will include the comprehensive results, allowing readers to compare and analyze the impact of removing "irrelevant" rules on the overall performance of our proposed model. We can see from Table 7, that the model without single-source initialization assigns relatively lower scores to rules, and there are still some less precise rules present in the Top-k rules (denoted in bold). Similarly, other ablated models exhibit less precise rules in the top-k results in Tables 8 and 9. This serves as evidence of the effectiveness of the full-version model.

Table 7: Some relations and their top-3 relevant relations in the full initialization REST. The relations are taken from WN18RR.

| Rule Head | Rule Body | Scores |
|---|---|---|
| _hypernym | _also_see-> _synset_domain_topic_of-> _derivationally_related_form | 0.66 |
|  | _verb_group-> _similar_to | 0.59 |
|  | _member_meronym-> _similar_to | 0.45 |
| _derivationally_related_form | _also_see-> _instance_hypernym | 0.74 |
|  | _derivationally_related_form-> _also_see | 0.68 |
|  | _has_part-> _has_part | 0.62 |
| _member_meronym | _has_part-> _hypernym | 0.62 |
|  | _has_part-> _instance_hypernym | 0.55 |
|  | _has_part-> _member_meronym-> _similar_to | 0.50 |
| _synset_domain_topic_of | _synset_domain_topic_of$^{-1}$-> _member_meronym-> _similar_to | 0.88 |
|  | _synset_domain_topic_of$^{-1}$-> _similar_to | 0.84 |
|  | _member_meronym-> _synset_domain_topic_of$^{-1}$ | 0.81 |

Table 8: Some relations and their top-3 relevant relations when using SUM as message function. The relations are taken from WN18RR.

| Rule Head | Rule Body | Score |
|---|---|---|
| _hypernym | _similar_to-> _hypernym$^{-1}$ | 0.93 |
|  | _similar_to-> _member_meronym | 0.90 |
|  | _hypernym$^{-1}$-> _also_see | 0.88 |
| _derivationally_related_form | _has_part-> _similar_to | 0.89 |
|  | _derivationally_related_form$^{-1}$-> _also_see | 0.88 |
|  | _synset_domain_topic_of-> _member_meronym-> _similar_to | 0.88 |
| _member_meronym | _member_meronym-> _similar_to | 0.90 |
|  | _has_part-> _synset_domain_topic_of | 0.85 |
|  | _instance_hypernym-> _derivationally_related_form-> _member_meronym | 0.85 |
| _synset_domain_topic_of | _also_see-> _has_part | 0.95 |
|  | _synset_domain_topic_of$^{-1}$-> _also_see | 0.95 |
|  | _hypernym-> _derivationally_related_form-> _also_see | 0.92 |

Table 9: Some relations and their top-3 relevant relations when using MUL as message function. The relations are taken from WN18RR.

| Rule Head | Rule Body | Score |
|---|---|---|
| _hypernym | _similar_to -> _hypernym$^{-1}$ | 0.93 |
|  | _instance_hypernym-> _instance_hypernym | 0.91 |
|  | _instance_hypernym-> _hypernym | 0.91 |
| _derivationally_related_form | _similar_to -> _derivationally_related_form | 0.94 |
|  | _derivationally_related_form-> _similar_to | 0.93 |
|  | _has_part-> _derivationally_related_form$^{-1}$ | 0.93 |
| _member_meronym | _member_meronym-> _also_see | 0.93 |
|  | _hypernym-> _similar_to | 0.90 |
|  | _similar_to-> _synset_domain_topic_of-> _derivationally_related_form | 0.87 |
| _synset_domain_topic_of | _also_see-> _verb_group | 0.92 |
|  | _synset_domain_topic_of$^{-1}$-> _similar_to | 0.86 |
|  | _has_part-> _synset_domain_topic_of$^{-1}$ | 0.84 |

## F  Sensitivity Analysis

We further conduct a sensitivity analysis of the subgraph hop $n$, where different $n$ represents the maximum number of neighbors extracted by REST within the subgraphs. We conduct experiments on both the same distribution (transductive setting) and different distribution benchmarks (inductive setting). As Tables 10, 11, 12, and 13 show, REST exhibit great robustness across different $n$.

Table 10: Experiments in the transductive setting with n = 2, 3, 4, where training and inference are in the same KG.

| Hop Number | wn_v1 | wn_v2 | wn_v3 | wn_v4 | fb_v1 | fb_v2 | fb_v3 | fb_v4 | nell_v1 | nell_v2 | nell_v3 | nell_v4 |
|---|---|---|---|---|---|---|---|---|---|---|---|---|
| 2hop | 84.65 | 85.40 | 87.66 | 85.73 | 84.35 | 91.78 | 95.67 | 96.97 | 92.48 | 96.59 | 95.73 | 93.89 |
| 3hop | 92.02 | 90.90 | 91.59 | 91.63 | 88.01 | 93.98 | 96.43 | 97.71 | 93.62 | 95.76 | 92.79 | 92.47 |
| 4hop | 89.97 | 89.96 | 90.19 | 89.68 | 88.74 | 95.12 | 96.16 | 97.44 | 94.99 | 96.64 | 96.26 | 94.35 |

Table 11: Experiments in the inductive setting on WN18RR benchmark with n = 2, 3, 4, where training and inference are in the different KGs.

| Hop Number | wn_v1_ind | wn_v2_ind | wn_v3_ind | wn_v4_ind |
|---|---|---|---|---|
| 2 hop Neighbors | 92.55 | 89.58 | 76.27 | 89.67 |
| 3 hop Neighbors | 96.28 | 94.56 | 79.50 | 94.19 |
| 4 hop Neighbors | 94.70 | 93.17 | 77.19 | 92.68 |

Table 12: Experiments in the inductive setting on FB15k237 with n = 2, 3, 4, where training and inference are in the different KGs.

| Hop Number | fb_v1_ind | fb_v2_ind | fb_v3_ind | fb_v4_ind |
|---|---|---|---|---|
| 2 hop Neighbors | 69.53 | 87.47 | 90.55 | 93.68 |
| 3 hop Neighbors | 75.12 | 91.21 | 93.06 | 96.06 |
| 4 hop Neighbors | 70.44 | 90.12 | 93.45 | 94.59 |

## G  More fine-grained metrics on the inductive benchmarks.

As Hits@10 scores reach a fairly high level, we provide the experimental results on more difficult and comprehensive metrics than Hits@10. Tables 14, 15, and 16 summarize the results of GraIL and our REST on MRR, Hits@1, and Hits@5. These representative metrics suggest that there is still a large room for improvement.

## H  More Baseline Results and Analysis

We provide additional baselines and our REST for comparison to obtain a more comprehensive explanation. These baselines include NBFNET [27], RMPI [40], and NODEPIECE [41]. The comparative results are presented in the following Table 17. REST also outperforms RMPI in a large margin and achieves competitive results compared with NodePiece and NBFNet. This implies the great potential for the subgraph-based methods to achieve superior results than the whole-graph-based methods. And we will focus on this point as our furture work.

Table 13: Experiments in the inductive setting on NELL995 with n = 2, 3, 4, where training and inference are in the different KGs.

| Hop Number | nell_v1_ind | nell_v2_ind | nell_v3_ind | nell_v4_ind |
|---|---|---|---|---|
| 2 hop Neighbors | 83.00 | 90.18 | 93.18 | 90.81 |
| 3 hop Neighbors | 88.00 | 94.96 | 96.79 | 92.61 |
| 4 hop Neighbors | 89.00 | 94.94 | 95.67 | 91.54 |

Table 14: MRR results on inductive benchmarks.

| Model | wn_v1 | wn_v2 | wn_v3 | wn_v4 | fb_v1 | fb_v2 | fb_v3 | fb_v4 | nell_v1 | nell_v2 | nell_v3 | nell_v4 |
|---|---|---|---|---|---|---|---|---|---|---|---|---|
| GraIL | 74.09 | 79.32 | 54.25 | 73.34 | 45.91 | 61.78 | 62.79 | 65.09 | 44.82 | 64.49 | 70.73 | 60.45 |
| REST | 81.08 | 87.06 | 62.66 | 88.14 | 49.71 | 65.74 | 68.75 | 72.41 | 55.99 | 69.65 | 74.99 | 69.63 |

Table 15: H@1 results on inductive benchmarks.

| Model | wn_v1 | wn_v2 | wn_v3 | wn_v4 | fb_v1 | fb_v2 | fb_v3 | fb_v4 | nell_v1 | nell_v2 | nell_v3 | nell_v4 |
|---|---|---|---|---|---|---|---|---|---|---|---|---|
| GraIL | 68.89 | 71.46 | 51.24 | 70.59 | 37.31 | 50.94 | 52.89 | 54.04 | 39.00 | 54.98 | 58.71 | 49.27 |
| REST | 71.28 | 82.54 | 54.88 | 84.89 | 39.05 | 53.14 | 56.42 | 60.39 | 43.00 | 56.72 | 61.80 | 57.59 |

Table 16: H@5 results on inductive benchmarks.

| Model | wn_v1 | wn_v2 | wn_v3 | wn_v4 | fb_v1 | fb_v2 | fb_v3 | fb_v4 | nell_v1 | nell_v2 | nell_v3 | nell_v4 |
|---|---|---|---|---|---|---|---|---|---|---|---|---|
| GraIL | 80.78 | 75.63 | 55.74 | 72.37 | 53.91 | 75.21 | 73.70 | 78.06 | 47.50 | 81.41 | 87.08 | 59.39 |
| REST | 92.55 | 90.02 | 69.59 | 89.99 | 60.49 | 80.75 | 84.28 | 87.64 | 57.00 | 86.97 | 92.21 | 86.32 |

Table 17: More inductive baseline results on the inductive benchmark.

| Model | wn_v1 | wn_v2 | wn_v3 | wn_v4 | fb_v1 | fb_v2 | fb_v3 | fb_v4 | nell_v1 | nell_v2 | nell_v3 | nell_v4 |
|---|---|---|---|---|---|---|---|---|---|---|---|---|
| RMPI | 89.63 | 83.22 | 73.14 | 81.42 | 71.71 | 83.37 | 86.01 | 88.69 | 60.50 | 94.01 | 95.36 | 87.62 |
| NBFNet | 94.80 | 90.50 | **89.30** | 89.00 | 83.40 | **94.90** | **95.10** | 96.00 | - | - | - | - |
| NodePiece | 83.00 | 88.60 | 78.50 | 80.70 | **87.30** | 93.90 | 94.40 | 94.90 | **89.00** | 90.10 | 93.60 | 89.30 |
| REST(**ours**) | **96.28** | **94.56** | 79.50 | **94.19** | 75.12 | 91.21 | 93.06 | **96.06** | 88.00 | **94.96** | **96.79** | **92.61** |

