# OpenReview forum: "Learning Rule-Induced Subgraph Representations for Inductive Relation Prediction"
_NeurIPS.cc/2023/Conference — NeurIPS 2023 poster_

### Official Review · Reviewer_s9SP · 2023-06-30

**Soundness:** 4 excellent
**Presentation:** 3 good
**Contribution:** 3 good
**Rating:** 6
**Confidence:** 5

**Summary:**

This paper proposes REST, a method for inductive link prediction over knowledge graphs. Compared to previous techniques that extract subgraphs around the target edge, REST avoids node labelling (thus saving substantial execution time) and only initializes the representation of the target link, setting other edge representations to zero vectors. Moreover, REST introduces edge-wise message passing to update edge representations iteratively, and achieves this through a series of learnable functions and a gated recurrent unit (GRU). Node updates are then computed using a long-short term memory (LSTM) cell, conditioned on the target query.

By only initializing the target edge representation, the paper shows that REST can focus only on "relevant" rules, i.e., rules including the target edge, thus substantially reducing noise. More precisely, REST is shown to learn rule-induced subgraph representations. Finally, on the empirical side, the paper runs experiments on standard baselines for inductive link prediction, achieving strong results, reports ablation studies highlighting the value of its initialization and recurrent components, and discusses further experiments showcasing candidate rule scoring by the learned REST model on WN18RR and efficiency gains versus other baselines.

**Strengths:**

- The paper is well-written and easy to follow.
- The design choices in REST (single-source initialization, message passing setup, etc.) are well-justified  and supported empirically.
- The paper provides a good overview of the literature in the related work section.
- The experimental results on inductive datasets are strong and convincing. The ablation studies also demonstrate the value of the REST model.
- The additional case study is very interesting.

**Weaknesses:**

- The paper does not report all baseline results in the experimental section (Table 1). For instance, NBFNet is cited in the related work section but not reported in the tables. From a quick glance at that paper, its results would indicate NBFNet has mostly superior performance on FB15k-237. Therefore, it would be important to include these results for completeness and ideally also discuss potential explanations for performance gains/losses.

- I think the experimental section can be improved by including further benchmarks from other link prediction setups. For instance, it would be interesting to see REST's performance on the transductive versions of the current benchmarks, and on OGB [1] benchmarks for link prediction.

[1] Hu et al. Open Graph Benchmark: Datasets for Machine Learning on Graphs, NeurIPS 2020.

**Questions:**

- Could you provide rule scores on WN18RR (similarly to the scores you currently have in the experimental section) with the ablated model without single-source? I think contrasting the scores and/or the different top-k rules across the full version and the ablated version would really help drive the point across regarding ignoring "irrelevant" rules.

**Limitations:**

The authors adequately discuss model limitations, particularly in terms of relatively poor scalability.

---

> ### Author Rebuttal · Authors · 2023-08-09
>
> We thank the reviewer for the positive and insightful comments. We respond to each comment as follows and sincerely hope that our rebuttal could properly address your concerns. If so, we would deeply appreciate it if you could raise your score. If not, please let us know your further concerns, and we will continue actively responding to your comments and improving our submission.
>
> **1. Including more baseline and analyze results on FB15k-237 datasets.**
>
> Thanks for pointing out this. We have included the baseline in the revised paper accordingly. We analyzed NBFNet's performance on FB15k-237 and understood the concerns raised regarding our model's comparative performance.
>
> Our analysis observed that FB15k-237 is a dataset with diverse relations but relatively smaller nodes. As such, it may contain more complex rules governing entity relationships. We hypothesize that models like NBFNet, which utilize a relatively larger number of negative samples, can better discriminate the correctness of intricate rules. This may potentially lead to improved performance of these models on FB15k-237.
>
> On the other hand, our model is particularly well-suited for sparse knowledge graphs with larger numbers of nodes. Our state-of-the-art results on datasets like WN18RR and NELL995 demonstrate that subgraph-based models tend to excel in reasoning over such knowledge graphs.
>
> **2. Further benchmarks of REST**
>
> We agree that evaluating REST's performance on transductive versions of the current benchmarks is crucial for comprehensively assessing its capabilities. As such, we have already conducted experiments in Table 13 on the transductive versions of the benchmarks and included the results in our revised paper.
>
> On the other hand, we acknowledge that the OGB benchmarks are also essential for evaluating link prediction models, and we regret that we were unable to include them in the initial submission due to time constraints. Nevertheless, we will include results on OGB benchmarks in the revised paper.
>
> Table 13: experiments the transductive versions of the current benchmarks. (Hits@10)
> |Model|wn_v1|wn_v2|wn_v3|wn_v4|fb_v1|fb_v2|fb_v3|fb_v4|nell_v1|nell_v2|nell_v3|nell_v4|
> |-|-|-|-|-|-|-|-|-|-|-|-|-|
> |GraIL|65.59|69.36|64.63|67.28|71.93|86.30|88.95|91.55|64.08|86.88|84.19|82.33|
> |RuleN|63.42|68.09|63.05|65.55|67.53|88.00|91.47|92.35|62.82|82.82|80.72|58.84|
> |REST|92.02|90.90|91.59|91.63|88.01|93.98|96.43|97.71|93.62|95.76|92.79|92.47|
>
> **3. Rule scores on WN18RR with the ablated model without single-source and other ablated models.**
>
> We have conducted additional experiments to obtain the rule scores on WN18RR with the ablated model without single-source and other ablated models. In the revised paper, we will include the comprehensive results, allowing readers to compare and analyze the impact of removing "irrelevant" rules on the overall performance of our proposed model.
>
> We can see from Table 14, that the model without single-source initialization assigns relatively lower scores to rules, and there are still some less precise rules present in the Top-k rules (denoted in **bold**). Similarly, other ablated models exhibit less precise rules in the top-k results in Tables 15 and 16. This serves as evidence of the effectiveness of the full-version model.
>
> Table 14: Some relations and their top-3 relevant relations when using full initialization method. The relations are taken from WN18RR.
> |Rule Head|Rule Body|Score|
> |-|-|-|
> |_hypernym|**_also_see->_synset_domain_topic_of->_derivationally_related_form**|**0.66**|
> ||_verb_group->_similar_to|0.59|
> ||**_member_meronym->_similar_to**|**0.45**|
> |_derivationally_related_form|_also_see->_instance_hypernym|0.74|
> ||**_derivationally_related_form->_also_see**|**0.68**|
> ||_has_part->_has_part|0.62|
> |_member_meronym|**_has_part->_hypernym**|**0.62**|
> ||**_has_part->_instance_hypernym**|**0.55**|
> ||_has_part->_member_meronym->_similar_to|0.50|
> |_synset_domain_topic_of|_synset_domain_topic_of^{-1}->_member_meronym->_similar_to|0.88|
> ||_synset_domain_topic_of^{-1}->_similar_to|0.84|
> ||_member_meronym->_synset_domain_topic_of^{-1}|0.81|
>
> Table 15: Some relations and their top-3 relevant relations when using SUM as message function. The relations are taken from WN18RR.
> |Rule Head|Rule Body|Score|
> |-|-|-|
> |_hypernym|_similar_to->_hypernym$^{-1}$|0.93|
> ||**_similar_to->_member_meronym**|**0.90**|
> ||_hypernym$^{-1}$->_also_see|0.88|
> |_derivationally_related_form|_has_part->_similar_to|0.89|
> ||_derivationally_related_form$^{-1}$->_also_see|0.88|
> ||**_synset_domain_topic_of->_member_meronym->_similar_to**|**0.88**|
> |_member_meronym|_member_meronym->_similar_to|0.90|
> ||_has_part->_synset_domain_topic_of|0.85|
> ||_instance_hypernym->_derivationally_related_form->_member_meronym|0.85|
> |_synset_domain_topic_of|_also_see->_has_part|0.95|
> ||_synset_domain_topic_of$^{-1}$->_also_see|0.95|
> ||_hypernym->_derivationally_related_form->_also_see|0.92|
>
> Table 16: Some relations and their top-3 relevant relations when using MUL as message function. The relations are taken from WN18RR.
> |Rule Head|Rule Body|Score|
> |-|-|-|
> |_hypernym|_similar_to->_hypernym$^{-1}$|0.93|
> ||**_instance_hypernym->_instance_hypernym**|**0.91**|
> ||**_instance_hypernym->_hypernym**|**0.91**|
> |_derivationally_related_form|**_similar_to->_derivationally_related_form**|**0.94**|
> ||_derivationally_related_form->_similar_to|0.93|
> ||_has_part->_derivationally_related_form$^{-1}$|0.93|
> |_member_meronym|_member_meronym->_also_see|0.93|
> ||**_hypernym->_similar_to**|**0.90**|
> ||_similar_to->_synset_domain_topic_of->_derivationally_related_form|0.87|
> |_synset_domain_topic_of|_also_see->_verb_group|0.92|
> ||_synset_domain_topic_of$^{-1}$->_similar_to|0.86|
> ||_has_part->_synset_domain_topic_of$^{-1}$|0.84|

---

> > ### Comment · Reviewer_s9SP · 2023-08-17
> > **Reviewer Response**
> >
> > Thanks very much for these experiments! I still have some concerns about the comparison with other baselines, but really appreciate the efforts you have made to improve your work. Overall, I will keep my initial verdict.

---

> > > ### Author Response · Authors · 2023-08-17
> > >
> > > Dear Reviewer s9SP,
> > >
> > > Thank you for acknowledging the improvements we've made based on your feedback.
> > >
> > > We sincerely appreciate your time and thoughtful evaluation of our work. Any further suggestions before the deadline are welcomed.
> > >
> > > Best regards,
> > > Authors

---

### Official Review · Reviewer_eBJJ · 2023-07-06

**Soundness:** 3 good
**Presentation:** 3 good
**Contribution:** 2 fair
**Rating:** 4
**Confidence:** 4

**Summary:**

This paper proposes a subgraph-based method to address the inductive relation prediction task, where the model learns to predict links by learning implicit rules from subgraphs. Models proposed in previous works lack the ability to differentiable the relevant and irrelevant paths. To address this, the authors propose several modifications including single-source initialization, edge-wise message passing, and modified activation functions. In the experiments, these modifications are shown to be effective which leads to superior performance compared to several subgraph-based and rule-based methods.

**Strengths:**

- The proposed single-source initialization and edge-wise message passing are well-motivated and well-presented.
- The paper is well-written and easy to read.


**Weaknesses:**

## Novelty


The authors propose several modifications to the standard message passing algorithm to better support inductive rule learning. This includes single-source initialization, edge-wise message passing, and modified activation functions. While these modifications are well-motivated and sensible, they are straightforward and mostly incremental changes to the standard protocol. That said the intellectual novelty is limited.

Section 5 provides an interesting theoretical analysis of the proposed model. While it is always nice to have a solid analysis, it does not contribute to any theoretical findings nor does it reveal novel properties of the proposed method other than proving it will work, which is already empirically proven in many previous works.

In summary, this paper proposes several incremental modifications to GNN-based inductive reasoning methods. While the modifications are rather straightforward, it does seem to lead to better empirical results.


## Quality


The proposed single-source initialization and edge-wise message passing are well-motivated and well-presented.

My main concern is more about the subgraph-based methods as a whole. This class of methods requires first extracting the subgraphs from the KG before they can be learned by the GNN models. And what concerns me the most is that the extraction step is not learnable: one has to pick a magic n-hop number beforehand and proceed to extract the subgraphs and finally train the models on them. This leads to several limitations: (1) it cannot generalize to KGs that have longer relational path than those seen in the training set; (2) the n-hop number are typically small, as the complexity of running GNN grows quickly with respect to the size of the graph. These limitations are non-existent for backward-chaining rule-based methods.

I understand this criticism is probably beyond the scope of this work, but I highly recommend the authors step outside of the "well-established" zone and challenge themselves to address these limitations. For example, as a first step, one can investigate what is the effect of the number n on the performance of the model on datasets with the same data distribution; and then on ones with different distributions, and if the performance deteriorates, propose solutions to alleviate that.


## Clarity

The paper is well-written and easy to read.


## Significance

The proposed method has some empirical significance as it is shown to outperform all previous baselines. However, as mentioned above, the model largely resembles the ones in prior work and only several incremental changes are proposed. More importantly, this class of methods has a serious limitation that is still left unaddressed in this work, which limits its applications in real-world tasks.

**Questions:**

None

**Limitations:**

Yes

---

> ### Author Rebuttal · Authors · 2023-08-09
>
> We thank the reviewer for the positive and insightful comments. We respond to each comment as follows and sincerely hope that our rebuttal could properly address your concerns. If so, we would deeply appreciate it if you could raise your score ("4: Borderline reject"). If not, please let us know your further concerns, and we will continue actively responding to your comments and improving our submission.
>
> ### Novelty:
> **1. The intellectual novelty is limited.**
>
> In this paper, we propose a single-source initialization and an edge-wise message passing, which effectively captures relevant rules and significantly increases the subgraph extraction efficiency. This architecture is well-motivated and intrinsically different from the standard protocol. The key idea behind the proposed method is to learn representations of **links** rather than **nodes**.
>
> Learning node representations brings two limitations in the standard protocol such as GraIL,
> 1. they need **node labeling** in the stage of subgraph preprocessing, leading to the major time cost during inference.
> 2. they need a **pooling layer** to get final subgraph representations, which incurs information loss and decreases the model performance.
>
> However, as shown in Figure 3, REST does not require node labeling and directly learns the representation of the target link as the subgraph representation. This significantly accelerates the subgraph preprocessing and improves the model performance.
>
> **2. The theoretical analysis does not contribute to any theoretical findings.**
>
> We apologize for any confusion in the original theoretical analysis.
>
> As stated in Lines 212 to 214, our theorem 2 suggests why the existing baseline GraIL and a series of its following works achieve suboptimal results. This is because massive irrelevant rules within the subgraph will make the model prone to overfitting while misleading the model to incorrect reasoning results. Empirical proof of this conclusion is that GraIL and other subgraph-based methods can get an AUC-PR of over 0.999 during training, while they can only get an AUC-PR of 0.900 during validation.
>
> Meanwhile, our theoretical analysis also provides insights for other IRP methods.
> - First, eliminating noises within the extracted subgraph is crucial for subgraph-based methods. While existing methods focus on data level to extract ad-hoc subgraphs, our model proposes a simple way for denoising at the model level, i.e., single-source edge-wise message passing.
> - Second, labeling tricks such as single-source initialization can effectively improve the model performance.
> - Last but not least, the idea of learning links is especially important in IRP task, as links play a vital role in reasoning.
>
> ### Quality:
> 1. **Subgraph-based methods cannot generalize to KGs that have longer relational path than those seen in the training set.**
>
> We admit that the model cannot generalize to KGs that have longer relational path than those in the training set. However, unclosing subgraphs wtih $n=3$ can capture paths at max length $l=6$. As shown in Figure 3 of Neural LP [1], when the number of paths length increases, the model performance decreases rapidly, especially with $l\geq 6$. Therefore, $n=3$ is enough for most scenarios.
>
> 2. **The n-hop number are typically small, as the complexity of running GNN grows quickly with respect to the size of the graph.**
>
> We agree that this is ubiquitous in existing subgraph-based methods. However, during inference, the largest time cost is subgraph preprocessing, i.e., subgraph extraction. To address this issue, we have attempted to remove node labeling during subgraph extraction and achieve relatively great efficiency. Our future works will continue to focus on handling this problem and making subgraph-based methods more applicable.
>
> 3. **One can investigate what is the effect of the number n on the performance of the model on datasets with the same data distribution; and then on ones with different distributions, and if the performance deteriorates, propose solutions to alleviate that.**
>
> Thanks for the valuable suggestions. We conduct experiments to further investigate the effect of $n$.
> For the same data distribution, we conduct experiments on the same KG (transductive setting) as follows:
>
> Table 11: experiments in the transductive setting with n = 2, 3, 4, where training and inference are in the same KG. (Hits@10)
> |Hop Number|wn_v1|wn_v2|wn_v3|wn_v4|fb_v1|fb_v2|fb_v3|fb_v4|nell_v1|nell_v2|nell_v3|nell_v4|
> |-|-|-|-|-|-|-|-|-|-|-|-|-|
> |2hop|84.65|85.40|87.66|85.73|84.35|91.78|95.67|96.97|92.48|96.59|95.73|93.89|
> |3hop|92.02|90.90|91.59|91.63|88.01|93.98|96.43|97.71|93.62|95.76|92.79|92.47|
> |4hop|89.97|89.96|90.19|89.68|88.74|95.12|96.16|97.44|94.99|96.64|96.26|94.35|
>
> For different data distributions, we conduct experiments on different KGs (inductive setting) as follows:
>
> Table 12: experiments in the inductive setting with n = 2, 3, 4, where training and inference are in the different KGs. (Hits@10)
> |Hop Number|wn_v1_ind|wn_v2_ind|wn_v3_ind|wn_v4_ind|fb_v1_ind|fb_v2_ind|fb_v3_ind|fb_v4_ind|nell_v1_ind|nell_v2_ind|nell_v3_ind|nell_v4_ind|
> |-|-|-|-|-|-|-|-|-|-|-|-|-|
> |2hop|92.55|89.58|76.27|89.67|69.53|87.47|90.55|93.68|83.00|90.18|93.18|90.81|
> |3hop|96.28|94.56|79.50|94.19|75.12|91.21|93.06|96.06|88.00|94.96|96.79|92.61|
> |4hop|94.70|93.17|77.19|92.68|70.44|90.12|93.45|94.59|89.00|94.94|95.67|91.54|
>
> From the results, we can find that in both transductive and inductive settings, $n=3$ achieves almost the best results. Increasing the hop number to 4 can only bring marginal improvements (e.g., results on fb_v1 and nell_v2), and usually results in model performance degradation (e.g., wn_v1_ind and fb_v1_ind). These results demonstrate our conclusion that $n=3$ is robust enough for most situations.
>
> [1] Yang, Fan, Zhilin Yang, and William W. Cohen. "Differentiable learning of logical rules for knowledge base reasoning." Advances in neural information processing systems 30 (2017).

---

> > ### Comment · Reviewer_eBJJ · 2023-08-20
> > **Thanks for the response**
> >
> > Thanks for the response.
> >
> > **Novelty**. I appreciate the clarification, however, my initial criticism is more about REST vs standard GNN instead of vs GRAIL, also node labeling is originally a design choice picked by GRAIL rather than a hard requirement for GNN-based models, and removing this dependency is not necessarily a challenging task.
> >
> > **Theoretical analysis and ablation study**. I appreciate the clarification and results and I recommend including them in the draft as well.
> >
> > **Subgraph-based methods**. Still, my main concern remains as this family of methods just simply cannot generalize to test data as other methods do, due to the fixed subgraph extraction. The authors also admitted it and stated that "a specific magic number is enough for most scenarios".
> >
> > That said, I will keep my score. However, I understand my main concern is not particular to this specific method, and addressing such limitation requires modifying the methodology fundamentally, which can lie beyond the scope of this work, and I will not be frustrated if it gets accepted.

---

> > > ### Author Response · Authors · 2023-08-21
> > >
> > > Thanks for the valuable and insightful comments.
> > >
> > > Overall, the main concerns of Reviewer eBJJ lies in the class of subgraph-based methods, and we would like to clarify the necessity of subgraph-based methods.
> > >
> > > First of all, a key limitation of standard GNN such as NBFNet [1] is the **scalability**. These methods rely on message passing on the whole KG to accelerate training and inference. During one message passing, these standard GNN models compute messages for all edges in the KG with a complexity of $O(\mathcal{E})$, which is unaffordable for large-scale KGs. This requirement severely limits the application of standard GNN models as they cannot generalize to large KGs. However, subgraph-based methods have the potential (though currently with a low efficiency) to reason on large-scale KGs, as the input is a subgraph rather than the whole graph. Existing method [2] on homogeneous graphs exhibits the efficiency potential of subgraph-based methods.
> > >
> > > Second, for the problem **"Subgraph-based methods cannot generalize to KGs that have longer relational path than those seen in the training set"**, this problem intrinsically lies in all the GNN-based methods. GNN-based methods with $k$ layers cannot generalize to KGs that have longer relational path with length $l> k$.
> > >
> > > In conclusion, we firmly believe that subgraph-based methods have some unique advantages against standard GNN models. We warmly welcome the reviewers to raise new questions if time permits. We hope our rebuttal can address the concerns raised by the reviewers. Thank the reviewer once again for taking the time to review our work and provide valuable feedback.
> > >
> > >
> > >
> > > [1] Zhu, Zhaocheng, et al. "Neural bellman-ford networks: A general graph neural network framework for link prediction." *Advances in Neural Information Processing Systems* 34 (2021): 29476-29490.
> > >
> > > [2] Yin, Haoteng, et al. "Algorithm and system co-design for efficient subgraph-based graph representation learning." *arXiv preprint arXiv:2202.13538* (2022).

---

### Official Review · Reviewer_r6aB · 2023-07-06

**Soundness:** 3 good
**Presentation:** 3 good
**Contribution:** 3 good
**Rating:** 6
**Confidence:** 3

**Summary:**

 This paper tackles a problem in the sub-area of machine learning on graphs. It looks at approaches for inductive reasoning (i.e. ones the generalize to unseen vertices), identifies a problem with those approaches and suggests a novel manner to tackle is problem, exploiting the recurrant nature of message-passing (via a GRU and LSTM).

There where theorems, along with proofs, that analized the potential effectiveness of the model. Moreover, the model was trained on multiple data sets and compared to multiple strong baselines. Finally, ablation studies where performed to empirically show the effect of the GRU and LSTM components.

The paper is overall clear to read.  Section 3 however could be easier to read through if the mathematical expressions where better separated from the text.

The field of inductive reasoning over graphs itself tackles a very relevant problem for machine learning on graphs. This paper identifies a problem with current approaches and introduces a fairly novel, effective and scalable approach taking this problem into account. While this is likely not be interesting to everyone in the world of machine learning, it should be interesting to many people in the machine learning on graphs community.



**Strengths:**

This paper is sound. To applies to its analysis of the problem at hand, but also to its experimental setup, which incluedes good baselines and ablation studies.


**Weaknesses:**

The mathematical nature of the paper, combined with a reliance on an understanding of several other things (e.g. subgraph representation, LSTMs and GRUs) could make this a difficult paper to get into for people from other sub-areas.

**Questions:**

With some scores being higher than 95%, do you think research in this direction should look into a more difficult task instead?


**Limitations:**

The reported hits@10 are fairly high (typically 90%-95%). It would have been nice if hits at a lower k would have been reported as well.

---

> ### Author Rebuttal · Authors · 2023-08-08
>
> We thank the reviewer for the positive and insightful comments. We respond to each comment as follows and sincerely hope that our rebuttal could properly address your concerns. If so, we would deeply appreciate it if you could raise your score. If not, please let us know your further concerns, and we will continue actively responding to your comments and improving our submission.
>
> **1. Section 3 could be easier to read if the mathematical expressions were better separated from the text.**
>
> Thanks for the detailed suggestion. We will reorganize Section 3 (i.e., problem definition) accordingly to improve the presentation of mathematical expressions in the revised paper.
>
> **2. The mathematical nature, with a reliance on an understanding of subgraph representation, LSTM and GRUs could make it difficult to get into for people from other sub-areas.**
>
> We will add annotations and improve the notations to make it easier to understand the math and core concepts, such as the subgraph representation of the model.
>
> **3. With some scores being higher than 95%, research in this direction should look into a more difficult task instead.  Hits at a lower k would have been reported as well.**
>
> As Hits@10 scores reach a fairly high level, we provide the experimental results on more difficult and comprehensive metrics than Hits@10. Tables 8, 9, and 10 summarize the results of GraIL and our REST on MRR, Hits@1, and Hits@5. These representative metrics suggest that there is still a large room for improvement. The evaluations on three datasets show that the proposed model significantly outperforms GraIL on all metrics. This demonstrates the effectiveness of our REST.
>
> Table 8: MRR results on inductive benchmarks.
> |Model|wn_v1|wn_v2|wn_v3|wn_v4|fb_v1|fb_v2|fb_v3|fb_v4|nell_v1|nell_v2|nell_v3|nell_v4|
> |-----|-----|-----|-----|-----|-----|-----|-----|-----|-------|-------|-------|-------|
> |GraIL|74.09|79.32|54.25|73.34|45.91|61.78|62.79|65.09|44.82|64.49|70.73|60.45|
> |REST|81.08|87.06|62.66|88.14|49.71|65.74|68.75|72.41|55.99|69.65|74.99|69.63|
>
> Table 9: H@1 results on inductive benchmarks.
> |Model|wn_v1|wn_v2|wn_v3|wn_v4|fb_v1|fb_v2|fb_v3|fb_v4|nell_v1|nell_v2|nell_v3|nell_v4|
> |-----|-----|-----|-----|-----|-----|-----|-----|-----|-------|-------|-------|-------|
> |GraIL|68.89|71.46|51.24|70.59|37.31|50.94|52.89|54.04|39.00|54.98|58.71|49.27|
> |REST|71.28|82.54|54.88|84.89|39.05|53.14|56.42|60.39|43.00|56.72|61.80|57.59|
>
> Table 10: H@5 results on inductive benchmarks.
> |Model|wn_v1|wn_v2|wn_v3|wn_v4|fb_v1|fb_v2|fb_v3|fb_v4|nell_v1|nell_v2|nell_v3|nell_v4|
> |-----|-----|-----|-----|-----|-----|-----|-----|-----|-------|-------|-------|-------|
> |GraIL|80.78|75.63|55.74|72.37|53.91|75.21|73.70|78.06|47.50|81.41|87.08|59.39|
> |REST|92.55|90.02|69.59|89.99|60.49|80.75|84.28|87.64|57.00|86.97|92.21|86.32|

---

### Official Review · Reviewer_UyGM · 2023-07-07

**Soundness:** 3 good
**Presentation:** 4 excellent
**Contribution:** 3 good
**Rating:** 7
**Confidence:** 3

**Summary:**

This work proposes a single-source edge-wise GNN model for the inductive relation prediction task, called REST. The method includes a single-source initialization approach that reduce the irrelevant rules during learning and an RNN-based edge-wise message passing mechanism to model the sequential property of mined rules. Authors experiments REST on three inductive relation prediction benchmarks and REST achieves  good results. And without node labeling, the subgraph extraction is significantly faster than GraIL.

**Strengths:**

The paper is well-written and easy to follow. The proposed method is well-motivated and is easy to understand. The experiments proves the effectiveness of the REST, especially the ablation study proves the effectiveness of the main novelty in the method, the single-source initialization and RNN-based edge-wise massage passing.

**Weaknesses:**

Some of the recent proposed inductive relation prediction methods are missing, such as
* Mingyang Chen, Wen Zhang, Yushan Zhu, Hongting Zhou, Zonggang Yuan, Changliang Xu, Huajun Chen: Meta-Knowledge Transfer for Inductive Knowledge Graph Embedding. SIGIR 2022: 927-937


**Questions:**

1. During experiment, is the binary operators in Equation (2) set the same as Theorem 1?
2. As shown in Table 2, replacing the GRU massage function to SUM and MUL significantly downgrade the performance which is more significant than change the single initialization to full initialization. Is there any reason?

**Limitations:**

See weakness and limitations.

---

> ### Author Rebuttal · Authors · 2023-08-08
>
> We thank the reviewer for the positive and insightful comments. We respond to each comment as follows and sincerely hope that our rebuttal could properly address your concerns. If so, we would deeply appreciate it if you could raise your score. If not, please let us know your further concerns, and we will continue actively responding to your comments and improving our submission.
>
> **1. Missing related works in paper.**
>
> Thank you for your valuable suggestions. We appreciate your insightful comments on our paper.
> In the revised manuscript, we have included the missing reference to the work by Mingyang Chen et al. [1], titled "Meta-Knowledge Transfer for Inductive Knowledge Graph Embedding."
> The reference to the mentioned paper has been added in the related work section to provide readers with a more comprehensive overview of recent advancements in inductive relation prediction methods.
>
> **2.  During experiment, is the binary operators in Equation (2) set the same as Theorem 1?**
>
> Regarding the binary operators in Equation (2) and Theorem 1, we apologize for any confusion in the original manuscript. In Equation (2), the binary operators are described as generalized operators, which can be replaced with various alternatives.
> During the experiment, we replaced the binary operators with the RNN-based functions we proposed.
>
> **3.  As shown in Table 2, replacing the GRU massage function to SUM and MUL significantly downgrade the performance which is more significant than change the single initialization to full initialization. Is there any reason?**
>
> The significant performance downgradation when using SUM and MUL instead of the GRU message function lies in the importance of position information in our edge-wise message passing framework. Unlike some previous works like GraIL, where node labeling implicitly encodes the position information, our approach learns position information explicitly through the GRU message function.
> We conducted case studies in the ablation experiments for SUM and MUL to further investigate the impact. Our findings reveal that the model captures some less precise cycles(denoted in **bold**) without the position information, leading to suboptimal results.
>
> Table 6: Some relations and their top-3 relevant relations when using SUM as message function. The relations are taken from WN18RR.
> |Rule Head|Rule Body|Score|
> |----------------------------|------------------------------------------------------------|--------|
> |_hypernym|_similar_to->_hypernym$^{-1}$|0.93|
> ||**_similar_to->_member_meronym**|**0.90**|
> ||_hypernym$^{-1}$->_also_see|0.88|
> |_derivationally_related_form|_has_part->_similar_to|0.89|
> ||_derivationally_related_form$^{-1}$->_also_see|0.88|
> ||**_synset_domain_topic_of->_member_meronym->_similar_to**|**0.88**|
> |_member_meronym|_member_meronym->_similar_to|0.90|
> ||_has_part->_synset_domain_topic_of|0.85|
> ||_instance_hypernym->_derivationally_related_form->_member_meronym|0.85|
> |_synset_domain_topic_of|_also_see->_has_part|0.95|
> ||_synset_domain_topic_of$^{-1}$->_also_see|0.95|
> ||_hypernym->_derivationally_related_form->_also_see|0.92|
>
> Table 7: Some relations and their top-3 relevant relations when using MUL as message function. The relations are taken from WN18RR.
> |Rule Head|Rule Body|Score|
> |----------------------------|------------------------------------------------------------|--------|
> |_hypernym|_similar_to->_hypernym$^{-1}$|0.93|
> ||**_instance_hypernym->_instance_hypernym**|**0.91**|
> ||**_instance_hypernym->_hypernym**|**0.91**|
> |_derivationally_related_form|**_similar_to->_derivationally_related_form**|**0.94**|
> ||_derivationally_related_form->_similar_to|0.93|
> ||_has_part->_derivationally_related_form$^{-1}$|0.93|
> |_member_meronym|_member_meronym->_also_see|0.93|
> ||**_hypernym->_similar_to**|**0.90**|
> ||_similar_to->_synset_domain_topic_of->_derivationally_related_form|0.87|
> |_synset_domain_topic_of|_also_see->_verb_group|0.92|
> ||_synset_domain_topic_of$^{-1}$->_similar_to|0.86|
> ||_has_part->_synset_domain_topic_of$^{-1}$|0.84|
>
> [1] Mingyang Chen, Wen Zhang, Yushan Zhu, Hongting Zhou, Zonggang Yuan, Changliang Xu,
> Huajun Chen: Meta-Knowledge Transfer for Inductive Knowledge Graph Embedding. SIGIR
> 2022: 927-937

---

> > ### Comment · Reviewer_UyGM · 2023-08-21
> > **Thanks for the response**
> >
> > Thanks the authors to clarify that the position information is important for edge-wise message passing network. Overall I think the single-source plus edge-wise message passing network is interesting. It gets ride of the work to find/design proper features for node initialization in the GNN and proves the GNN still works without carefully node feature design. I would like to keep my initial score.

---

> > > ### Author Response · Authors · 2023-08-21
> > >
> > > Dear Reviewer UyGM,
> > >
> > > Thank you for acknowledging the improvements we've made based on your feedback.
> > >
> > > We sincerely appreciate your time and thoughtful evaluation of our work. Any further suggestions before the deadline are welcomed.
> > >
> > > Best regards, Authors

---

### Author Rebuttal · Authors · 2023-08-08

We thank all the reviewers for the insightful comments and constructive suggestions, which are very helpful for us to strengthen this submission.

We have submitted detailed responses to address the concerns raised by all the reviewers. Besides answering the technical questions,
1. we conduct more case studies for each ablated model, which demonstrates the effectiveness of each component within proposed REST; (for reviewers UyGM and s9SP)
2. we conduct experiments in both transductive and inductive settings while providing more results to show the effect of the hop number $n$; (for reviewers eBJJ and s9SP)
3. we provide more results with other comprehensive metrics, such as MRR, Hits@1, and Hits@5; (for reviewer r6aB)
4. we clarify the advantages of proposed method and give our theoretical findings, which demonstrates the intellectual novelty of our REST. (for reviewer eBJJ)

We would like to know if our responses have properly addressed your concerns. All of your feedback and/or additional comments are warmly welcomed.

---

### Decision · Program_Chairs · 2023-09-21

**Decision:**

Accept (poster)

**Comment:**

Previous  inductive link prediction approaches extract subgraphs around the target edge. They lack the ability to differentiate the relevant and irrelevant paths. This paper proposes to improve previous approaches by single-source initialization, edge-wise message passing, and modified activation functions.

By only initializing the target edge representation, the paper shows that REST can focus only on "relevant" rules, i.e., rules including the target edge, thus substantially reducing noise. More precisely, REST is shown to learn rule-induced subgraph representations.
Experiments on standard baselines for inductive link prediction shows strong results. Ablation studies highlighting the value of its initialization and recurrent components, and  efficiency gains versus other baselines.

Strengths:
1. The paper is well-written and easy to follow.
2. The design choices (single-source initialization, message passing setup, etc.) are well-justified and supported empirically.
3. The paper provides a good overview of the literature in the related work section.
4. The experimental results on inductive datasets are strong.
5. The ablation studies are informative

Weaknesses:
1. The paper can be stronger by including other stronger baselines and further benchmarks.